# Adaptive Universal Generalized PageRank Graph Neural Network

**Eli Chien & Jianhao Peng**[*]
Department of Electrical and Computer Engineering
University of Illinois Urbana-Champaign, USA
{ichien3,jianhao2}@illinois.edu

**Pan Li**
Department of Computer Science
Purdue University, USA
panli@purdue.edu

**Olgica Milenkovic**
Department of Electrical and Computer Engineering
University of Illinois Urbana-Champaign, USA
milenkov@illinois.edu

## Abstract

In many important graph data processing applications the acquired information includes both node features and observations of the graph topology. Graph neural networks (GNNs) are designed to exploit both sources of evidence but they do not optimally trade-off their utility and integrate them in a manner that is also universal. Here, universality refers to independence on homophily or heterophily graph assumptions. We address these issues by introducing a new Generalized PageRank (GPR) GNN architecture that adaptively learns the GPR weights so as to jointly optimize node feature and topological information extraction, regardless of the extent to which the node labels are homophilic or heterophilic. Learned GPR weights automatically adjust to the node label pattern, irrelevant on the type of initialization, and thereby guarantee excellent learning performance for label patterns that are usually hard to handle. Furthermore, they allow one to avoid feature over-smoothing, a process which renders feature information nondiscriminative, without requiring the network to be shallow. Our accompanying theoretical analysis of the GPR-GNN method is facilitated by novel synthetic benchmark datasets generated by the so-called contextual stochastic block model. We also compare the performance of our GNN architecture with that of several state-of-the-art GNNs on the problem of node-classification, using well-known benchmark homophilic and heterophilic datasets. The results demonstrate that GPR-GNN offers significant performance improvement compared to existing techniques on both synthetic and benchmark data. Our implementation is available online.[1]

## 1 Introduction

Graph-centered machine learning has received significant interest in recent years due to the ubiquity of graph-structured data and its importance in solving numerous real-world problems such as semi-supervised node classification and graph classification (Zhu, 2005; Shervashidze et al., 2011; Lü & Zhou, 2011). Usually, the data at hand contains two sources of information: Node features and graph topology. As an example, in social networks, nodes represent users that have different combinations of interests and properties captured by their corresponding feature vectors; edges on the other hand document observable friendship and collaboration relations that may or may not depend on the node features. Hence, learning methods that are able to simultaneously and adaptively exploit node features and the graph topology are highly desirable as they make use of their latent connections and thereby improve learning on graphs.

Graph neural networks (GNN) leverage their representational power to provide state-of-the-art performance when addressing the above described application domains. Many GNNs use message

---

[*]equal contribution
[1]https://github.com/jianhao2016/GPRGNN

passing (Gilmer et al., 2017; Battaglia et al., 2018) to manipulate node features and graph topology. They are constructed by stacking (graph) neural network layers which essentially propagate and transform node features over the given graph topology. Different types of layers have been proposed and used in practice, including graph convolutional layers (GCN) (Bruna et al., 2014; Kipf & Welling, 2017), graph attention layers (GAT) (Velickovic et al., 2018) and many others (Hamilton et al., 2017; Wijesinghe & Wang, 2019; Zeng et al., 2020; Abu-El-Haija et al., 2019).

However, most of the existing GNN architectures have two fundamental weaknesses which restrict their learning ability on general graph-structured data. First, most of them seem to be tailor-made to work on *homophilic (associative) graphs*. The homophily principle (McPherson et al., 2001) in the context of node classification asserts that nodes from the same class tend to form edges. Homophily is also a common assumption in graph clustering (Von Luxburg, 2007; Tsourakakis, 2015; Dau & Milenkovic, 2017) and in many GNNs design (Klicpera et al., 2018). Methods developed for homophilic graphs are nonuniversal in so far that they fail to properly solve learning problems on *heterophilic (disassortative) graphs* (Pei et al., 2019; Bojchevski et al., 2019; 2020). In heterophilic graphs, nodes with distinct labels are more likely to link together (For example, many people tend to preferentially connect with people of the opposite sex in dating graphs, different classes of amino acids are more likely to connect within many protein structures (Zhu et al., 2020) etc). GNNs model the homophily principle by aggregating node features within graph neighborhoods. For this purpose, they use different mechanisms such as averaging in each network layer. Neighborhood aggregation is problematic and significantly more difficult for heterophilic graphs (Jia & Benson, 2020).

Second, most of the existing GNNs fail to be "deep enough". Although in principle an arbitrary number of layers may be stacked, practical models are usually shallow (including 2-4 layers) as these architectures are known to achieve better empirical performance than deep networks. A widely accepted explanation for the performance degradation of GNNs with increasing depth is *feature-over-smoothing,* which may be intuitively explained as follows. The process of GNN feature propagating represents a form of random walks on "feature graphs," and under proper conditions, such random walks converge with exponential rate to their stationary points. This essentially levels the expressive power of the features and renders them nondiscriminative. This intuitive reasoning was first described for linear settings in Li et al. (2018) and has been recently studied in Oono & Suzuki (2020) for a setting involving nonlinear rectifiers.

We address these two described weaknesses by combining GNNs with Generalized PageRank techniques (GPR) within a new model termed GPR-GNN. The GPR-GNN architecture is designed to first learn the hidden features and then to propagate them via GPR techniques. The focal component of the network is the GPR procedure that associates each step of feature propagation with a *learnable weight*. The weights depend on the contributions of different steps during the information propagation procedure, and they can be both positive and negative. This departures from common nonnegativity assumptions (Klicpera et al., 2018) allows for the signs of the weights to adapt to the homophily/heterophily structure of the underlying graphs. The amplitudes of the weights trade-off the degree of smoothing of node features and the aggregation power of topological features. These traits do not change with the choice of the initialization procedure and elucidate the process used to combine node features and the graph structure so as to achieve (near)-optimal predictions. In summary, the GPR-GNN method can simultaneously learn the node label patterns of disparate classes of graphs and prevent feature over-smoothing.

The excellent performance of GPR-GNN is demonstrated empirically, on real world datasets, and further supported through a number of theoretical findings. In the latter setting, we show that the GPR procedure relates to general polynomial graph filtering, which can naturally deal with both high and low frequency parts of the graph signals. In contrast, recent GNN models that utilize Personalized PageRanks (PPR) with fixed weights (Wu et al., 2019; Klicpera et al., 2018; 2019) inevitably act as low-pass filters. Thus, they fail to learn the labels of heterophilic graphs. We also establish that GPR-GNN can provably mitigate the feature-over-smoothing issue in an adaptive manner even after large-step propagation (i.e., after a large number of propagation steps). Hence, the method is able to make use of informative large-step propagation.

To test the performance of GPR-GNN on homophilic and heterophilic node label patterns and determine the trade-off between node and topological feature exploration, we first describe the recently proposed *contextual stochastic block model* (cSBM) (Deshpande et al., 2018). The cSBM allows for smoothly controlling the "informativeness ratio" between node features and graph topology,

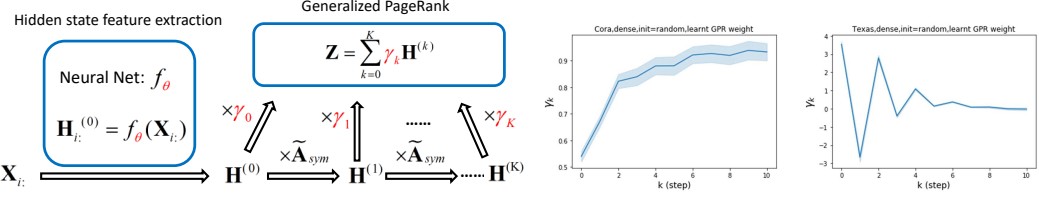

(a) Illustration of our proposed GPR-GNN model.  (b) Cora, $\mathcal{H}(G) = 0.656$ (c) Texas, $\mathcal{H}(G) = 0.016$

Figure 1: (a) Hidden state feature extraction is performed by a neural networks using individual node features propagated via GPR. Note that both the GPR weights $\gamma_k$ and parameter set $\{\theta\}$ of the neural network are learned simultaneously in an end-to-end fashion (as indicated in red). (b)-(c) The learnt GPR weights of the GPR-GNN on real world datasets. Cora is homophilic while Texas is heterophilic (Here, $\mathcal{H}$ stands for the level of homophily defined below). An interesting trend may be observed: For the heterophilic case the weights alternate from positive to negative with dampening amplitudes (more examples are provided in Section 5). The shaded region corresponds to a 95% confidence interval.

where the graph can vary from being highly homophilic to highly heterophilic. We show that GPR-GNN outperforms all other baseline methods for the task of semi-supervised node classification on the cSBM consistently from strong homophily to strong heterophily. We then proceed to show that GPR-GNN offers state-of-the-art performance on node-classification benchmark real-world datasets which contain both homophilic and heterophilic graphs. Due to the space limit, we put all proofs, formal theorem statements, and the conclusion section in the Supplement.

## 2 PRELIMINARIES

Let $G = (V, E)$ be an undirected graph with nodes $V$ and edges $E$. Let $n$ denote the number of nodes, assumed to belong to one of $C \geq 2$ classes. The nodes are associated with the node feature matrix $\mathbf{X} \in \mathbb{R}^{n \times f}$, where $f$ denotes the number of features per node. Throughout the paper, we use $\mathbf{X}_{i:}$ to indicate the $i^{\text{th}}$ row and $\mathbf{X}_{:j}$ to indicate the $j^{\text{th}}$ column of the matrix $\mathbf{X}$, respectively. The symbol $\delta_{ij}$ is reserved for the Kronecker delta function. The graph $G$ is described by the adjacency matrix $\mathbf{A}$, while $\tilde{\mathbf{A}}$ stands for the adjacency matrix for a graph with added self-loops. We let $\tilde{\mathbf{D}}$ be the diagonal degree matrix of $\tilde{\mathbf{A}}$ and $\tilde{\mathbf{A}}_{\text{sym}} = \tilde{\mathbf{D}}^{-1/2} \tilde{\mathbf{A}} \tilde{\mathbf{D}}^{-1/2}$ denote the symmetric normalized adjacency matrix with self-loops.

## 3 GPR-GNNS: MOTIVATION AND CONTRIBUTIONS

**Generalized PageRanks.** Generalized PageRank (GPR) methods were first used in the context of unsupervised graph clustering where they showed significant performance improvements over Personalized PageRank (Kloumann et al., 2017; Li et al., 2019). The operational principles of GPRs can be succinctly described as follows. Given a seed node $s \in V$ in some cluster of the graph, a one-dimensional feature vector $\mathbf{H}^{(0)} \in \mathbb{R}^{n \times 1}$ is initialized according to $\mathbf{H}^{(0)}_{v:} = \delta_{vs}$. The GPR score is defined as $\sum_{k=0}^{\infty} \gamma_k \tilde{\mathbf{A}}^k_{\text{sym}} \mathbf{H}^{(0)} = \sum_{k=0}^{\infty} \gamma_k \mathbf{H}^{(k)}$, where the parameters $\gamma_k \in \mathbb{R}$, $k = 0, 1, 2, \ldots$, are referred to as the GPR weights. Clustering of the graph is performed locally by thresholding the GPR score. Certain PangRank methods, such as Personalized PageRank or heat-kernel PageRank (Chung, 2007), are associated with specific choices of GPR weights (Li et al., 2019). For an excellent in-depth discussion of PageRank methods, the interested reader is referred to (Gleich, 2015). The work in Li et al. (2019) recently introduced and theoretically analyzed a special form of GPR termed *Inverse PR* (IPR) and showed that long random walk paths are more beneficial for clustering then previously assumed, provided that the GPR weights are properly selected (Note that IPR was developed for homophilic graphs and optimal GPR weights for heterophilic graphs are not currently known).

**Equivalence of the GPR method and polynomial graph filtering.** If we truncate the infinite sum in the definition of GPR at some natural number $K$, $\sum_{k=0}^{K} \gamma_k \tilde{\mathbf{A}}^k_{\text{sym}}$ corresponds to a polynomial

graph filter of order $K$. Thus, learning the optimal GPR weights is equivalent to learning the optimal polynomial graph filter. Note that one can approximate any graph filter using a polynomial graph filter (Shuman et al., 2013) and hence the GPR method is able to deal with a large range of different node label patterns. Also, increasing $K$ allows one to better approximate the underlying optimal graph filter. This once again shows that large-step propagation is beneficial.

**Universality with respect to node label patterns: Homophily versus heterophily.** In their recent work, Pei et al. (2019) proposed an index to measure the level of homophily of nodes in a graph $\mathcal{H}(G) = \frac{1}{|V|} \sum_{v \in V} \frac{\text{Number of neighbors of } v \in V \text{ that have the same label as } v}{\text{Number of neighbors of } v}$. Note that $\mathcal{H}(G) \to 1$ corresponds to strong homophily while $\mathcal{H}(G) \to 0$ indicates strong heterophily. Figures 1 (b) and (c) plot the GPR weights learnt by our GPR-GNN method on a homophilic (Cora) and heterophilic (Texas) dataset. The learnt GPR weights from Cora match the behavior of IPR (Li et al., 2019), which verifies that large-step propagation is indeed of great importance for homophilic graphs. The GPR weights learnt from Texas behave significantly differently from all known PR variants, taking a number of negative values. These differences in weight patterns are observed under random initialization, demonstrating that the weights are actually learned by the network and not forced by specific initialization. Furthermore, the large difference in the GPR weights for these two graph models illustrates the learning power of GPR-GNN and their universal adaptability.

**The over-smoothing problem.** One of the key components in most GNN models is the graph convolutional layer, described by

$$\mathbf{H}_{\text{GCN}}^{(k)} = \text{ReLU}\left( \tilde{\mathbf{A}}_{\text{sym}} \mathbf{H}_{\text{GCN}}^{(k-1)} \mathbf{W}^{(k)} \right), \ \hat{\mathbf{P}}_{\text{GCN}} = \text{softmax}\left( \tilde{\mathbf{A}}_{\text{sym}} \mathbf{H}_{\text{GCN}}^{(K-1)} \mathbf{W}^{(k)} \right),$$

where $\mathbf{H}_{\text{GCN}}^{(0)} = \mathbf{X}$ and $\mathbf{W}^{(k)}$ represents the trainable weight matrix for the $k^{\text{th}}$ layer. The key issue that limits stacking multiple layers is the over-smoothing phenomenon: If one were to remove ReLU in the above expression, $\lim_{k \to \infty} \tilde{\mathbf{A}}_{\text{sym}}^k \mathbf{H}^{(0)} = \mathbf{H}^{(\infty)}$, where each row of $\mathbf{H}^{(\infty)}$ only depends on the degree of the corresponding node, provided that the graph is irreducible and aperiodic. This shows that the model looses discriminative information provided by the node features as the number of layers increases.

**Mitigating graph heterophily and over-smoothing issues with the GPR-GNN model.** GPR-GNN first extracts hidden state features for each node and then uses GPR to propagate them. The GPR-GNN process can be mathematically described as:

$$\hat{\mathbf{P}} = \text{softmax}(\mathbf{Z}), \ \mathbf{Z} = \sum_{k=0}^{K} \gamma_k \mathbf{H}^{(k)}, \ \mathbf{H}^{(k)} = \tilde{\mathbf{A}}_{\text{sym}} \mathbf{H}^{(k-1)}, \ \mathbf{H}_{i:}^{(0)} = f_\theta(\mathbf{X}_{i:}), \tag{1}$$

where $f_\theta(.)$ represents a neural network with parameter set $\{\theta\}$ that generates the hidden state features $\mathbf{H}^{(0)}$. The GPR weights $\gamma_k$ are trained together with $\{\theta\}$ in an end-to-end fashion. The GPR-GNN model is easy to interpret: As already pointed out, GPR-GNN has the ability to adaptively control the contribution of each propagation step and adjust it to the node label pattern. Examining the learnt GPR weights also helps with elucidating the properties of the topological information of a graph (i.e., determining the optimal polynomial graph filter), as illustrated in Figure 1 (b) and (c).

**Placing GPR-GNNs in the context of related prior work.** Among the methods that differ from repeated stacking of GCN layers, APPNP (Klicpera et al., 2018) represents one of the state-of-the-art GNNs that is related to our GPR-GNN approach. It can be easily seen that APPNP as well as SGC (Wu et al., 2019) are special cases of our model since APPNP fixes $\gamma_k = \alpha(1-\alpha)^k, \gamma_K = (1-\alpha)^K$, while SGC removes all nonlinearities with $\gamma_k = \delta_{kK}$, respectively. These two weight choices correspond to Personalized PageRank (PPR) (Jeh & Widom, 2003), which is known to be suboptimal compared to the IPR framework when applied to homophilic node classification (Li et al., 2019). Fixing the GPR weights makes the model unable to adaptively learn the optimal propagation rules which is of crucial importance: As we will show in Section 4, the fixed PPR weights corresponds to low-pass graph filters which makes them inadequate for learning on heterophilic graphs. The recent work (Klicpera et al., 2018) showed that fixed PPR weights (APPNP) can also provably resolve the over-smoothing problem. However, the way APPNP prevents over-smoothing is independent on the node label information. In contrast, the escape of GPR-GNN from over-smoothing is guided by the node label information (Theorem 4.2). A detailed discussion of this phenomena along with illustrative examples is delegated to the Supplement.

Among the GCN-like models, JK-Net (Xu et al., 2018) exhibits some similarities with GPR-GNN. It also aggregates the outputs of different GCN layers to arrive at the final output. On the other hand, the GCN-Cheby method (Defferrard et al., 2016; Kipf & Welling, 2017) is related to polynomial graph filtering, where each convolutional layer propagates multiple steps and the graph filter is related to Chebyshev polynomials. In both cases, the depth of the models is limited in practice (Klicpera et al., 2018) and they are not easy to interpret as our GPR-GNN method. Some prior work also emphasizes adaptively learning the importance of different steps (Abu-El-Haija et al., 2018; Berberidis et al., 2018). Nevertheless, none of the above works is applicable for semi-supervised learning with GNNs and considers heterophilic graphs.

## 4 THEORETICAL PROPERTIES OF GPR-GNNS

**Graph filtering aspects of GPR-GNNs.** As mentioned in Section 3, the GPR component of the network may be viewed as a polynomial graph filter. Let $\tilde{\mathbf{A}}_{\text{sym}} = \mathbf{U}\mathbf{\Lambda}\mathbf{U}^T$ be the eigenvalue decomposition of $\tilde{\mathbf{A}}_{\text{sym}}$. Then, the corresponding polynomial graph filter equals $\sum_{k=0}^{K} \gamma_k \tilde{\mathbf{A}}_{\text{sym}}^k = \mathbf{U}g_{\gamma,K}(\mathbf{\Lambda})\mathbf{U}^T$, where $g_{\gamma,K}(\mathbf{\Lambda})$ is applied element-wise and $g_{\gamma,K}(\lambda) = \sum_{k=0}^{K} \gamma_k \lambda^k$. We established the following result.

**Theorem 4.1** (Informal). *Assume that the graph $G$ is connected. If $\gamma_k \geq 0$ $\forall k \in \{0, 1, ..., K\}$, $\sum_{k=0}^{K} \gamma_k = 1$ and $\exists k' > 0$ such that $\gamma_{k'} > 0$, then $g_{\gamma,K}(\cdot)$ is a low-pass graph filter. Also, if $\gamma_k = (-\alpha)^k, \alpha \in (0, 1)$ and $K$ is large enough, then $g_{\gamma,K}(\cdot)$ is a high-pass graph filter.*

By Theorem 4.1 and from our discussion in Section 3, we know that both APPNP and SGC will invariably suppress the high frequency components. Thus, they are inadequate for use on heterophilic graphs. In contrast, if one allows $\gamma_k$ to be negative and learned adaptively the graph filter will pass relevant high frequencies. This is what allows GPR-GNN to perform exceptionally well on heterophilic graphs (see Figure 2(c)).

**GPR-GNN can escape from over-smoothing.** As already emphasized, one crucial innovation of the GPR-GNN method is to make the GPR weights adaptively learnable, which allows GPR-GNN to avoid over-smoothing and trade node and topology feature informativeness. Intuitively, when large-step propagation is not beneficial, it increases the training loss. Hence, the corresponding GPR weights should decay in magnitude. This observation is captured by the following result, whose more formal statement and proof are delegated to the Supplement due to space limitations.

**Theorem 4.2** (Informal). *Assume the graph $G$ is connected and the training set contains nodes from each of the classes. Also assume that $k'$ is large enough so that the over-smoothing effect occurs for $\mathbf{H}^{(k)}, \forall k \geq k'$ which dominate the contribution to the final output $\mathbf{Z}$. Then, the gradients of $\gamma_k$ and $\gamma_k$ are identical in sign for all $k \geq k'$.*

Theorem 4.2 shows that as long as over-smoothing happens, $|\gamma_k|$ will approach 0 for all $k \geq k'$ when we use an optimizer such as stochastic gradient descent (SGD) which has a suitable learning rate decay. This reduces the contribution of the corresponding steps $\mathbf{H}^{(k)}$ in the final output $\mathbf{Z}$. When the weights $|\gamma_k|$ are small enough so that $\mathbf{H}^{(k)}$ no longer dominates the value of the final output $\mathbf{Z}$, the over-smoothing effect is eliminated.

## 5 RESULTS FOR NEW CSBM SYNTHETIC AND REAL-WORLD DATASETS

**Synthetic data.** In order to test the ability of label learning of GNNs on graphs with arbitrary levels of homophily and heterophily, we propose to use cSBMs (Deshpande et al., 2018) to generate synthetic graphs. We consider the case with two equal-size classes. In cSBMs, the node features are Gaussian random vectors, where the mean of the Gaussian depends on the community assignment. The difference of the means is controlled by a parameter $\mu$, while the difference of the edge densities in the communities and between the communities is controlled by a parameter $\lambda$. Hence $\mu$ and $\lambda$ capture the "relative informativeness" of node features and the graph topology, respectively. Moreover, positive $\lambda$'s correspond to homophilic graphs while negative $\lambda$'s correspond to heterophilic graphs. The information-theoretic limits of reconstruction for the cSBM are characterized in Deshpande et al. (2018). The results show that, asymptotically, one needs $\lambda^2 + \mu^2/\xi > 1$ to ensure a

vanishing ratio of the misclassified nodes and the total number of nodes, where $\xi = n/f$ and $f$ as before denotes the dimension of the node feature vector.

Note that given a tolerance value $\epsilon > 0$, $\lambda^2 + \mu^2/\xi = 1 + \epsilon$ is an arc of an ellipsoid for which $\lambda \geq 0$ and $\mu \geq 0$. To fairly and continuously control the extent of information carried by the node features and graph topology, we introduce a parameter $\phi = \arctan(\frac{\lambda\sqrt{\xi}}{\mu}) \times \frac{2}{\pi}$. The setting $\phi = 0$ indicates that only node features are informative, while $|\phi| = 1$ indicates that only the graph topology is informative. Moreover, $\phi = 1$ corresponds to strongly homophilic graphs while $\phi = -1$ corresponds to strongly heterophilic graphs. Note that the values $\phi$ and $-\phi$ convey the same amount of information regarding graph topology. This is due to the fact that $\lambda^2 = (-\lambda)^2$. Ideally, GNNs that are able to optimally learn on both homophilic and heterophilic graph should have similar performances for $\phi$ and $-\phi$. Due to space limitation we refer the interested reader to (Deshpande et al., 2018) for a review of all formal theoretical results and only outline the cSBM properties needed for our analysis. Additional information is also available in the Supplement.

Our experimental setup examines the semi-supervised node classification task in the transductive setting. We consider two different choices for the random split into training/validation/test samples, which we call sparse splitting (2.5%/2.5%/95%) and dense splitting (60%/20%/20%), respectively. The sparse splittnig is more similar to the original semi-supervised setting considered in Kipf & Welling (2017) while the dense setting is considered in Pei et al. (2019) for studying heterophilic graphs. We run each experiment 100 times with multiple random splits and different initializations.

*Methods used for comparisons.* We compare GPR-GNN with 6 baseline models: MLP, GCN (Kipf & Welling, 2017), GAT (Velickovic et al., 2018), JK-Net (Xu et al., 2018), GCN-Cheby (Defferrard et al., 2016), APPNP (Klicpera et al., 2018), SGC (Wu et al., 2019), SAGE (Hamilton et al., 2017) and Geom-GCN (Pei et al., 2019). For all architectures, we use the corresponding Pytorch Geometric library implementations (Fey & Lenssen, 2019). For Geom-GCN, we directly use the code provided by the authors[2]. We could not test Geom-GCN on cSBM and other datasets not originally tested in the paper due to a preprocessing subroutine that is not publicly available (Pei et al., 2019).

*The GPR-GNN model setup and hyperparameter tuning.* We choose random walk path lengths with $K = 10$ and use a 2-layer (MLP) with 64 hidden units for the NN component. For the GPR weights, we use different initializations including PPR with $\alpha \in \{0.1, 0.2, 0.5, 0.9\}$, $\gamma_k = \delta_{0k}$ or $\delta_{Kk}$ and the default random initialization in pytorch. Similarly, for APPNP we search the optimal $\alpha$ within $\{0.1, 0.2, 0.5, 0.9\}$. For other hyperparameter tuning, we optimize the learning rate over $\{0.002, 0.01, 0.05\}$ and weight decay $\{0.0, 0.0005\}$ for all models. For Geom-GCN, we use the best variants in the original paper for each dataset. Finally, we use GPR-GNN(rand) to describe the results obtained with random initialization of the GPR weights. Further experimental settings are discussed in the Supplement.

*Results.* We examine the robustness of all baseline methods and GPR-GNN using cSBM-generated data with $\phi \in \{-1, -0.75, -0.5, ..., 1\}$, which includes graphs across the heterophily/homophily spectrum. The results are summarized in Figure 2. For both the sparse and dense setting, GPR-GNN significantly outperforms all other baseline models whenever $\phi < 0$ (heterophilic graphs). On the other hand, all baseline GNNs can be worse then simple MLP when the graph information is weak ($\phi = 0, -0.25$). This shows that existing GNNs cannot apply to arbitrary graphs, while GPR-GNN is clearly more robust. APPNP methods have the worst performance on strongly heterophilic graphs. This is in agreement with the result of Theorem 4.1 which asserts that APPNP intrinsically acts a low-pass filter and is thus inadequate for strong heterophily settings. JKNet, GCN-Cheby and SAGE are the only three baseline models that are able to learn strongly heterophilic graphs under dense splitting. This is also to be expected since JKNet is the only baseline model that combines results from different steps at the last layer, which is similar to what is done in GPR-GNN. GCN-Cheby uses multiple steps in each layers which allows it to partially adapt to heterophilic settings as each layer is related to a polynomial graph filter of higher order compared to that of GCN. SAGE treats ego-embeddings and embeddings from neighboring nodes differently and does not simply average them out. This allows SAGE to adapt to the heterophilic case since the ego-embeddings prevent nodes from being overwhelmed by information from their neighbors. Nevertheless, JKNet, GCN-Cheby and SAGE are not deep in practice.

---

[2]https://github.com/graphdml-uiuc-jlu/geom-gcn

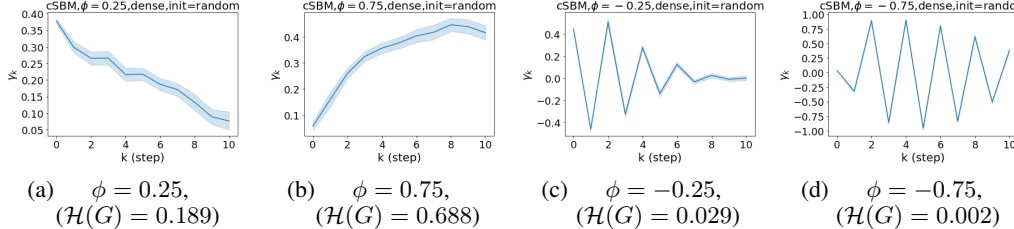

(a) $\phi = 0.25$, ($\mathcal{H}(G) = 0.189$)

(b) $\phi = 0.75$, ($\mathcal{H}(G) = 0.688$)

(c) $\phi = -0.25$, ($\mathcal{H}(G) = 0.029$)

(d) $\phi = -0.75$, ($\mathcal{H}(G) = 0.002$)

Figure 3: Figure (a)-(d) shows the learnt GPR weights by GPR-GNN with random initialization on cSBM, dense split. The shaded region indicates $95\%$ confidence interval.

Moreover, JKNet fails to learn under the sparse splitting model while GCN-Cheby and SAGE fail to learn well when the graph information is strong ($|\phi| \geq 0.5$), again under the sparse splitting model.

Also, we observe that random initialization of our GPR weights only results in slight performance drops under dense splitting. The drop is more evident for sparse splitting setting but our method still outperforms baseline models by a large margin for strongly heterophilic graphs. This is also to be expected as we have less label information in the sparse splitting setting where the implicit bias provided by good GPR initialization is helpful. The implicit bias becomes irrelevant for the dense splitting setting, since the label information is sufficiently rich.

Besides the strong performance of GPR-GNN, the other benefit is its interpretability. In Figure 3, we demonstrate the learnt GPR weights by our GPR-GNN on cSBM with random initialization. When the graph is weak homophilic ($\phi = 0.25$), the learnt GPR weights are decreasing. This is similar to the PPR weights used in APPNP, despite that the decaying speed is different. When the graph is strong homophilic ($\phi = 0.75$), the learnt GPR weights are increasing which is significantly different from the PPR weights. This result matches the recent finding in Li et al. (2019)

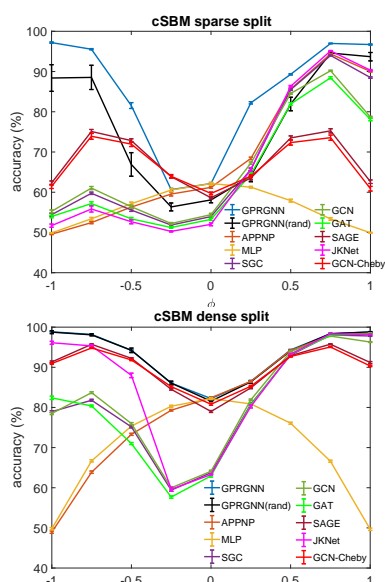

Figure 2: Accuracy of tested models on cSBM. Error bars indicate $95\%$ confidence interval.

and behave similar to IPR proposed by the authors. On the other hand, the learnt GPR weights have zig-zag shape when the graph is heterophilic. This again validates Theorem 4.1 as GPR weights with alternating signs correspond to a high-pass filter. Interestingly, when $\phi = -0.25$ the magnitude of learnt GPR weight is decreasing. This is because the graph information is weak and the node feature information is more important in this case. It makes sense that the learnt GPR weight focus on the first few steps. Hence, we have validated the interpretablity of GPR-GNN. In practice, one can use the learnt GPR weights to better understand the graph structured data at hand. We showcase this benefit in the results of real world benchmark datasets.

**Real world benchmark datasets.** We use 5 homophilic benchmark datasets available from the Pytorch Geometric library, including the citation graphs Cora, CiteSeer, PubMed (Sen et al., 2008; Yang et al., 2016) and the Amazon co-purchase graphs Computers and Photo (McAuley et al., 2015; Shchur et al., 2018). We also use 5 heterophilic benchmark datasets tested in Pei et al. (2019), including Wikipedia graphs Chameleon and Squirrel, the Actor co-occurrence graph, and webpage graphs Texas and Cornell from WebKB[3]. We summarize the dataset statistics in Table 1.

*Results on real-world datasets.* We use accuracy (the micro-F1 score) as the evaluation metric along with a $95\%$ confidence interval. The relevant results are summarized in Table 2. For homophilic datasets, we provide results for sparse splitting which is more aligned with the original setting used in Kipf & Welling (2017); Shchur et al. (2018). For the heterophilic datasets, we adopt dense splitting which is used in Pei et al. (2019).

---

[3]http://www.cs.cmu.edu/afs/cs.cmu.edu/project/theo-11/www/wwkb

Table 1: Benchmark dataset properties and statistics.

| Dataset | Cora | Citeseer | PubMed | Computers | Photo | Chameleon | Squirrel | Actor | Texas | Cornell |
|---|---|---|---|---|---|---|---|---|---|---|
| Classes | 7 | 6 | 5 | 10 | 8 | 5 | 5 | 5 | 5 | 5 |
| Features | 1433 | 3703 | 500 | 767 | 745 | 2325 | 2089 | 932 | 1703 | 1703 |
| Nodes | 2708 | 3327 | 19717 | 13752 | 7650 | 2277 | 5201 | 7600 | 183 | 183 |
| Edges | 5278 | 4552 | 44324 | 245861 | 119081 | 31371 | 198353 | 26659 | 279 | 277 |
| $\mathcal{H}(G)$ | 0.656 | 0.578 | 0.644 | 0.272 | 0.459 | 0.024 | 0.055 | 0.008 | 0.016 | 0.137 |

Table 2: Results on real world benchmark datasets: Mean accuracy (%) $\pm$ 95% confidence interval. Boldface letters are used to mark the best results while underlined boldface letters indicate results within the given confidence interval of the best result.

| | Cora | Citeseer | PubMed | Computers | Photo | Chameleon | Actor | Squirrel | Texas | Cornell |
|---|---|---|---|---|---|---|---|---|---|---|
| GPRGNN | **79.51**±**0.36** | 67.63±0.38 | **85.07**±**0.09** | **82.90**±**0.37** | **91.93**±**0.26** | **67.48**±**0.40** | **39.30**±**0.27** | **49.93**±**0.53** | **92.92**±**0.61** | **91.36**±**0.70** |
| APPNP | **79.41**±**0.38** | **68.59**±**0.30** | **85.02**±**0.09** | 81.99±0.26 | 91.11±0.26 | 51.91±0.56 | 38.86±0.24 | 34.77±0.34 | 91.18±0.70 | **91.80**±**0.63** |
| MLP | 50.34±0.48 | 52.88±0.51 | 80.57±0.12 | 70.48±0.28 | 78.69±0.30 | 46.72±0.46 | 38.58±0.25 | 31.28±0.27 | 92.26±0.71 | **91.36**±**0.70** |
| SGC | 70.81±0.67 | 58.98±0.47 | 82.09±0.11 | 76.27±0.36 | 83.80±0.46 | 63.02±0.43 | 29.39±0.20 | 43.14±0.28 | 55.18±1.17 | 47.80±1.50 |
| GCN | 75.21±0.38 | 67.30±0.35 | 84.27±0.01 | 82.52±0.32 | 90.54±0.21 | 60.96±0.78 | 30.59±0.23 | 45.66±0.39 | 75.16±0.96 | 66.72±1.37 |
| GAT | 76.70±0.42 | 67.20±0.46 | 83.28±0.12 | 81.95±0.38 | 90.09±0.27 | 63.9±0.46 | 35.98±0.23 | 42.72±0.33 | 78.87±0.86 | 76.00±1.01 |
| SAGE | 70.89±0.54 | 61.52±0.44 | 81.30±0.10 | **83.11**±**0.23** | 90.51±0.25 | 62.15±0.42 | 36.37±0.21 | 41.26±0.26 | 79.03±1.20 | 71.41±1.24 |
| JKNet | 73.22±0.64 | 60.85±0.76 | 82.91±0.11 | 77.80±0.97 | 87.70±0.70 | 62.92±0.49 | 33.41±0.25 | 44.72±0.48 | 75.53±1.16 | 66.73±1.73 |
| GCN-Cheby | 71.39±0.51 | 65.67±0.38 | 83.83±0.12 | 82.41±0.28 | 90.09±0.48 | 59.96±0.51 | 38.02±0.23 | 40.67±0.31 | 86.08±0.96 | 85.33±1.04 |
| GeomGCN | 20.37±1.13 | 20.30±0.90 | 58.20±1.23 | NA | NA | 61.06±0.49 | 31.81±0.24 | 38.28±0.27 | 58.56±1.77 | 55.59±1.59 |

Table 2 shows that, in general, GPR-GNN outperforms all tested methods. On homophilic datasets, GPR-GNN achieves the state-of-the-art performance. On heterophilic datasets, GPR-GNN significantly outperforms all the other baseline models. It is important to point out that there are two different patterns to be observed among the heterophilic datasets. On Chameleon and Squirrel, MLP and APPNP perform worse then other baseline methods such as GCN and JKNet. In contrast, MLP and APPNP outperform the other baseline methods on Actor, Texas and Cornell. We conjecture that this is due to the fact that the graph topology information is strong and weak, respectively. Note that these two patterns match the results of the cSBM experiments for $\phi$ close to $-1$ and 0, respectively (Figure 2). Furthermore, the homophily measure $\mathcal{H}(G)$ proposed by Pei et al. (2019) cannot characterize such differences in heterophilic datasets. We relegate the more detailed discussion of this topic along with illustrative examples to the Supplement. For fairness, we also repeated the experiment involving GeomGCN on homophilic datasets using a dense split - the observed performance pattern tends to be similar which can be found in Supplement.

We also examined the learned GPR weights on real datasets in Figure 4. Due to space limitations, a more comprehensive GPR weight analysis for other datasets is deferred to the Supplement. We can see that learned GPR weights are all positive for homophilic datasets (PubMed and Photo). In contrast, some GPR weights learned from heterophilic datasets (Actor and Squirrel) are negative. These results agree with the patterns observed on cSBMs. Interestingly, the learned weight $\gamma_0$ has the largest magnitude for the Actor dataset. This indicates that most of the information is contained in node features. From Table 2 we can also see that MLPs indeed outperforms most baseline GNNs (this is similar to the case of cSBM($\phi = -0.25$)). On the other hand, GPR weights learned from Squirrel have a zig-zag pattern. This implies that graph topology is more informative for Squirrel compared to Actor. From Table 2 we also see that baseline GNNs also outperform MLPs on Squirrel.

*Escaping from over-smoothing and dynamics of learning GPR weights.* To demonstrate the ability of GPR-GNNs to escape from over-smoothing, we choose the initial GPR weights to be $\gamma_k = \delta_{kK}$. This ensures that over-smoothing effects are present with high probability at the very beginning of the learning process. On cSBM($\phi = -1$) with dense splitting, we find that for 96 out of 100 runs, GPR-GNN predicts the same labels for all nodes at epoch 0, which implies that over-smoothing indeed occurs immediately. The final prediction is 98.79% accurate which is much larger than the initial accuracy of 50.07% at epoch 0. Similar results can be observed for other datasets and this verifies our theoretical findings. We plot the dynamics of the learned GPR weights in Figure 4(e)-(h), which shows that the peak at last step is indeed reduced while the GPR weights for other steps are significantly increased in magnitude. More results on the dynamics of learning GPR weights may be found in the Supplement.

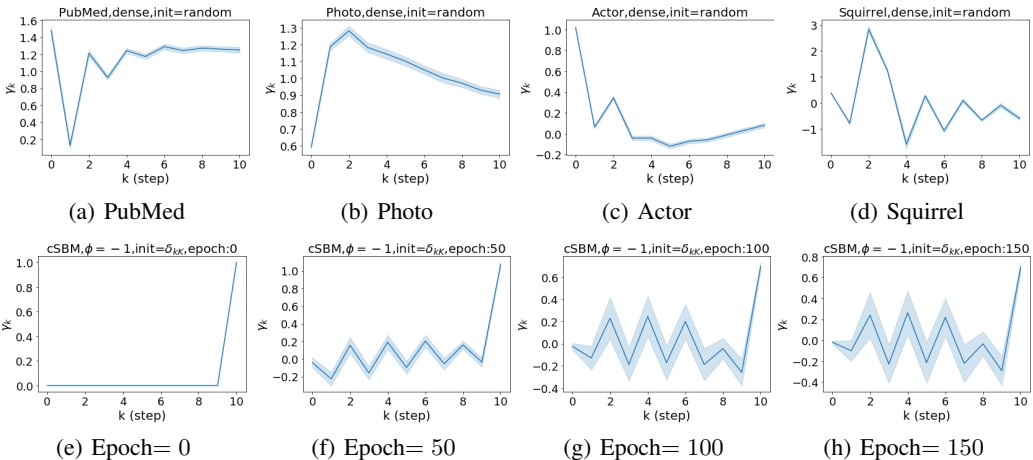

Figure 4: Figures (a)-(d) show the learned GPR weights of our GPR-GNN method with random initialization on various datasets, for dense splitting. Figures (e)-(f) show the learned weights of our GPR-GNN method with initialization $\delta_{kK}$ on cSBM($\phi = -1$), for dense splitting. The shaded region indicates a 95% confidence interval.

Table 3: Efficiency on selected real world benchmark datasets: Average running time per epoch(ms)/average total running time(s). Note that Geom-GCN requires a preprocessing procedure so we do not include it in the table. Complete efficiency table for all benchmark datasets is in Supplementary due to space limit.

|  | Cora | Pubmed | Computers | Chameleon | Actor | Squirrel | Texas |
|---|---|---|---|---|---|---|---|
| GPRGNN | 17.62ms / 3.74s | 20.19ms / 5.53s | 39.93ms / 11.40s | 16.74ms / 3.40s | 19.31ms / 4.49s | 25.28ms / 5.12s | 17.56ms / 3.55s |
| APPNP | 17.16ms / 4.00s | 18.47ms / 6.29s | 39.59ms / 20.00s | 17.01ms / 3.44s | 16.32ms / 4.04s | 22.93ms / 4.63s | 15.96ms / 3.24s |
| MLP | 4.14ms / 0.92s | 5.43ms / 2.86s | 5.33ms / 2.77s | 3.41ms / 0.69s | 4.84ms / 0.98s | 5.19ms / 1.05s | 3.81ms / 1.04s |
| SGC | 3.31ms / 3.31s | 3.81ms / 3.81s | 4.36ms / 4.36s | 3.13ms / 3.13s | 3.98ms / 1.00s | 4.79ms / 4.79s | 2.86ms / 2.09s |
| GCN | 9.25ms / 1.97s | 14.11ms / 4.17s | 32.45ms / 16.29s | 13.83ms / 2.79s | 12.39ms / 2.50s | 27.11ms / 5.56s | 10.22ms / 2.06s |
| GAT | 14.78ms / 3.42s | 21.52ms / 6.70s | 61.45ms / 24.28s | 16.63ms / 3.63s | 18.91ms / 3.86s | 47.46ms / 10.05s | 15.50ms / 3.13s |
| SAGE | 12.06ms / 2.44s | 28.82ms / 6.32s | 171.36ms / 71.94s | 64.43ms / 13.02s | 27.95ms / 5.65s | 343.47ms / 69.38s | 6.08ms / 1.28s |
| JKNet | 18.97ms / 4.41s | 24.48ms / 6.61s | 35.02ms / 14.96s | 20.03ms / 5.15s | 23.52ms / 4.75s | 29.89ms / 6.67s | 19.67ms / 4.01s |
| GCN-cheby | 22.96ms / 4.75s | 45.76ms / 12.02s | 218.82ms / 96.58s | 89.41ms / 18.06s | 43.94ms / 8.88s | 440.55ms / 88.99s | 12.34ms / 3.08s |

*Efficiency analysis.* We also examine the computational complexity of GPR-GNNs compared to other baseline models. We report the empirical training time in Table 3. Compared to APPNP, we only need to learn $K+1$ additional GPR weights for GPR-GNN, and usually $K \leq 20$ (i.e. we choose $K = 10$ in our experiments). This additional computations are dominated by the computations performed by the neural network module $f_\theta$. We can observe from Table 3 that indeed GPR-GNN has a running time similar to that of APPNP. It is nevertheless worth pointing out that the authors of Bojchevski et al. (2020) successfully scaled APPNP to operate on large graphs. Whether the same techniques may be used to scale GPR-GNNs is an interesting open question.

## 6 CONCLUSIONS

We addressed two fundamental weaknesses of existing GNNs: Failing to act as universal learners by not generalizing to heterophilic graphs and making use of large number of propagation steps. We developed a novel GPR-GNN architecture which combines adaptive generalized PageRank (GPR) scheme with GNNs. We theoretically showed that our method does not only mitigates feature over-smoothing but also works on highly diverse node label patterns. We also tested GPR-GNNs on both homophilic and heterophilic node label patterns, and proposed a novel synthetic benchmark datasets generated by the contextual stochastic block model. Our experiments on real-world benchmark datasets showed clear performance gains of GPR-GNN over the state-of-the-art methods. Moreover, we showed that GPR-GNN has desirable interpretability properties which is of independent interest.

ACKNOWLEDGMENTS

The work was supported in part by the NSF Emerging Frontiers of Science of Information Grant 0939370 and the NSF CIF 1618366 Grant.

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

# A APPENDIX

## A.1 DETAILED DISCUSSION ON PREVENTING OVER-SMOOTHING.

As mentioned in Section 4, another method – APPNP – can also provably prevents over-smoothing Klicpera et al. (2018). The authors of this study use the fact that the PPR propagation will converge to $\mathbf{\Pi}_{\text{ppr}}\mathbf{H}^{(0)}$, where $\mathbf{\Pi}_{\text{ppr}} = \alpha(\mathbf{I}_n - (1-\alpha)\tilde{\mathbf{A}}_{\text{sym}})^{-1}$ is independent on the node label information provided in the training data. Each row of $\mathbf{\Pi}_{\text{ppr}}\mathbf{H}^{(0)}$ still depends on $\mathbf{H}^{(0)}$ and thus APPNP will not suffer from the over-smoothing effect. However, since $\mathbf{\Pi}_{\text{ppr}}$ is independent of the label information, it can cause undesired consequences that we discuss in what follows.

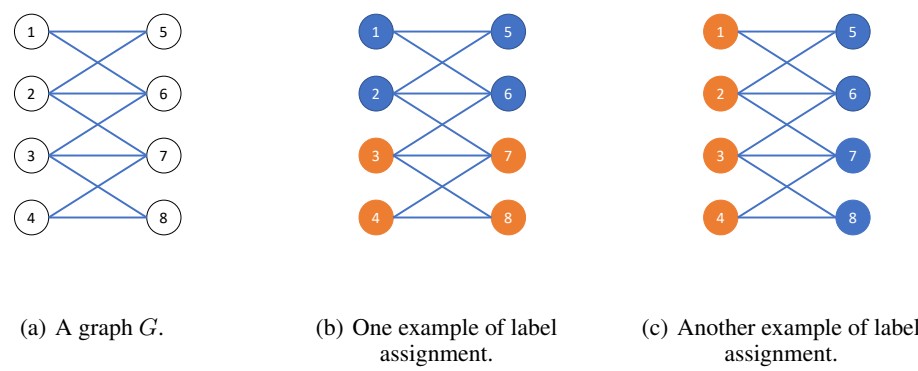

(a) A graph $G$.   (b) One example of label assignment.   (c) Another example of label assignment.

Figure 5: A simple example demonstrating how GPR-GNN escapes over-smoothing.

Let us consider a simple example shown in Figure 5 involving a connected and undirected graph $G = (V, E)$ (Figure 5 (a)). Consider two different node label assignments shown in Figure 5 (b) and Figure 5 (c). Obviously, the graph topologies depicted in Figure 5 (b) and (c) are identical and the only difference is the class label assignment. In Figure 5 (b), the graph is homophilic and hence the optimal graph filter should emphasize the low-frequency part of the graph signal. In contrast, in Figure 5 (c), the graph is heterophilic as the graph is bipartite with respect to the labels. Hence, the optimal graph filter should emphasize the high-frequency part of the graph signal. This example illustrates that the optimal graph filter should depend on both the graph topology and the node label information. Recall that the equivalent graph filter that APPNP uses in the asymptotic regime is $\mathbf{\Pi}_{\text{ppr}}$ which is independent on the node label information. Also, Theorem 4.1 established that APPNP intrinsically utilizes a low-pass filter. In contrast, GPR-GNN learns the GPR weights guided by the node label information which allows it to account for both cases (homophilic and heterophilic) shown.

## A.2 DISCUSSION ON THE INSUFFICIENCY OF HOMOPHILY MEASURE $\mathcal{H}(G)$

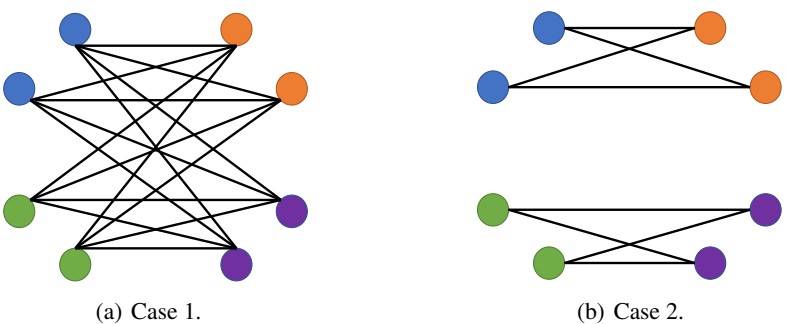

(a) Case 1.   (b) Case 2.

Figure 6: A simple example for explaining the insufficiency of homophily measure $\mathcal{H}(G)$.

As mentioned in Section 5, the homophily measure $\mathcal{H}(G)$ is inadequate for characterizing whether a heterophilic graph topology is informative or not. Consider two simple examples depicted in Fig-

ure 6, where the color of the nodes indicates their label. In case 1, blue and green nodes link to all orange and purple nodes. In case 2, blue nodes only link to orange nodes and green nodes only link to purple nodes. From the definition of $\mathcal{H}(G)$ one can see that both cases have $\mathcal{H}(G) = 0$, since in both cases nodes do not link to other nodes of the same label. However, it is obvious that the graph topology carries more node label information in case 2 compared to case 1. In fact, for case 1 it is impossible to distinguish blue and green nodes merely from the graph topology (and the same is true of orange and purple nodes). One possible alternative for the homophily measure is the Chernoff-Hellinger divergence Abbe (2017) of the empirical edge probability matrix $\mathbf{B}$; here $\mathbf{B}_{ij}$ is the empirical probability of an edge with one end node labeled $i$ and the other labeled $j$. The intuition behind our suggestion lies in the fact that the Chernoff-Hellinger divergence characterizes the fundamental limit of SBMs. However, as many practical graph generative processes may significantly differ from SBMs, investigating alternative homophily/heterophily measures is another interesting open problem.

### A.3 PROOF OF THEOREM 4.1

We first state the formal version of Theorem 4.1.

**Theorem A.1** (Formal version of Theorem 4.1)**.** *Assume the graph $G$ is connected. Let $\lambda_1 \geq \lambda_2 \geq ... \geq \lambda_n$ be the eigenvalues of $\tilde{\mathbf{A}}_{sym}$. If $\gamma_k \geq 0 \ \forall k \in \{0, 1, ..., K\}$, $\sum_{k=0}^{K} \gamma_k = 1$ and $\exists k' > 0$ such that $\gamma_{k'} > 0$, then $|g_{\gamma,K}(\lambda_i)/g_{\gamma,K}(\lambda_1)| < |\lambda_i/\lambda_1| \ \forall i \geq 2$. Also, if $\gamma_k = (-\alpha)^k, \alpha \in (0, 1)$ and $K \to \infty$, then $|\lim_{K\to\infty} g_{\gamma,K}(\lambda_i)/\lim_{K\to\infty} g_{\gamma,K}(\lambda_1)| > |\lambda_i/\lambda_1| \ \forall i \geq 2$.*

Note that $|g_{\gamma,K}(\lambda_i)/g_{\gamma,K}(\lambda_1)| < |\lambda_i/\lambda_1| \ \forall i \geq 2$ implies that after applying the graph filter $g_{\gamma,K}$, the lowest frequency component (correspond to $\lambda_1$) further dominates. Hence $g_{\gamma,K}$ acts like a low pass filter in this case. In contrast, $|\lim_{K\to\infty} g_{\gamma,K}(\lambda_i)/\lim_{K\to\infty} g_{\gamma,K}(\lambda_1)| > |\lambda_i/\lambda_1| \ \forall i \geq 2$ implies that after applying the graph filter, the lowest frequency component (correspond to $\lambda_1$) no longer dominates. This correspond to the high pass filter case.

*Proof.* We start with the low pass filter result. From basic spectral analysis (Von Luxburg, 2007) we know that $\lambda_1 = 1$ and $|\lambda_i| < 1, \forall i \geq 2$. One can also find the analysis in the proof of our Lemma A.2 in the Supplement. Then by assumption we know that

$$g_{\gamma,K}(\lambda_1) = \sum_{k=0}^{K} \gamma_k = 1.$$

Hence, proving Theorem A.1 is equivalent to show

$$|g_{\gamma,K}(\lambda_i)| < |\lambda_i| \ \forall i \geq 2.$$

This is obvious since $g_{\gamma,K}(\lambda) = \sum_{k=0}^{K} \gamma_k \lambda^k$ is a polynomial of order $K$ with nonnegative coefficients. It is easy to check that $\forall k \geq 1, \ |\lambda|^k < |\lambda|, \forall |\lambda| < 1$. Combine with the fact that all $\gamma_k$'s are nonnegative we have

$$|g_{\gamma,K}(\lambda_i)| \leq \sum_{k=0}^{K} \gamma_k |\lambda^k| = \sum_{k=0}^{K} \gamma_k |\lambda|^k \overset{(a)}{\leq} \sum_{k=0}^{K} \gamma_k |\lambda| = |\lambda|.$$

Finally, note that the only possibility that the inequality (a) holds is $\gamma_k = \delta_{0,K}$ since $\forall k \geq 1, \ |\lambda|^k < |\lambda|, \forall |\lambda| < 1$. However, by assumption $\sum_{k=0}^{K} \gamma_k = 1$ and $\exists k' > 0$ such that $\gamma_{k'} > 0$ we know that this is impossible. Hence (a) is a strict inequality $<$. Together we complete the proof for low pass filtering part.

For the high pass filter result, it is not hard to see that

$$\lim_{K\to\infty} g_{\gamma,K}(\lambda) = \lim_{K\to\infty} \sum_{k=0}^{K} \gamma_k \lambda^k = \lim_{K\to\infty} \sum_{k=0}^{K} (-\alpha\lambda)^k = \frac{1}{1 + \alpha\lambda},$$

where the last step is due to the fact that $\alpha \in (0, 1)$ and thus $\lim_{K\to\infty}(-\alpha\lambda)^K = 0, \forall |\lambda| \leq 1$. Thus we have

$$\left| \frac{\lim_{K\to\infty} g_{\gamma,K}(\lambda_i)}{\lim_{K\to\infty} g_{\gamma,K}(\lambda_1)} \right| = \left| \frac{1 + \alpha}{1 + \alpha\lambda_i} \right| \overset{(b)}{>} 1 \overset{(c)}{>} |\lambda_i| \ \forall i \geq 2.$$

Both strict inequalities (b) and (c) are from the fact that $|\lambda_i| < 1, \forall i \geq 2$. Notably, $\sup_{\lambda \in [1,-1)} \frac{1}{1+\alpha\lambda}$ happens at the boundary $\lambda = -1$, which corresponds the the bipartite graph. It further shows that the graph filter with respect to the choice $\gamma_k = (-\alpha)^k$ emphasizes high frequency components and thus it is indeed acting as a high pass filter. $\qquad\square$

## A.4 PROOF OF THEOREM 4.2

We start by introducing some additional notation, lemmas and definition before we proceed to the formal statement of Theorem 4.2. The label matrix is denoted by $\mathbf{Y} \in \mathbb{R}^{n \times C}$, where each row is a one-hot vector. We use $\mathbf{1}[\boldsymbol{\beta}] \in \mathbb{R}^C$ to denote the argmax of the vector $\boldsymbol{\beta} \in \mathbb{R}^C$: we have $\mathbf{1}[\boldsymbol{\beta}]_i = 1$ if and only if $\boldsymbol{\beta}_i = \max(\boldsymbol{\beta})$ (ties are broken evenly), and $\mathbf{1}[\boldsymbol{\beta}]_i = 0$ otherwise. Let us replace the softmax$(\cdot)$ with softmax$_\eta(\cdot)$, where we let softmax$_\eta(\boldsymbol{\beta})_i = e^{\eta\boldsymbol{\beta}_i}/(\sum_j e^{\eta\boldsymbol{\beta}_j})$ stand for the softmax with a smooth parameter $\eta > 0$. Note that for $\eta = 1$ we recover the standard softmax. With a slight abuse of notation, for the vector $\boldsymbol{\beta}$ we write $\exp(\boldsymbol{\beta})$ to denote element-wise exponentiation. We use $\langle \cdot, \cdot \rangle$ to denote the standard Euclidean inner product. Also we use $L$ for the cross entropy loss where

$$L = \sum_{i \in V} -\log(\langle \hat{\mathbf{P}}_{i:}, \mathbf{Y}_{i:} \rangle).$$

**Lemma A.2.** *Assume that the nodes in an undirected and connected graph $G$ have one of $C$ labels. Then, for $k$ large enough, we have*

$$\mathbf{H}_{:j}^{(k)} = \boldsymbol{\beta}_j \boldsymbol{\pi} + o_k(1) \; \forall j \in [C], \; \text{where } \boldsymbol{\pi}_i = \frac{\sqrt{\tilde{\mathbf{D}}_{ii}}}{\sqrt{\sum_{v \in V} \tilde{\mathbf{D}}_{vv}}} \; \text{and } \boldsymbol{\beta}^T = \boldsymbol{\pi}^T \mathbf{H}^{(0)}. \tag{2}$$

For any $\mathbf{H}^{(0)}$ and large enough $k \leq K$, if the label prediction is dominated by $\mathbf{H}^{(k)}$, all nodes will have a representation proportional to $\gamma_k \boldsymbol{\beta}$. Hence, we will arrive at the same label for all nodes. This is what we refer to as the over-smoothing phenomenon.

**Definition A.3** (The over-smoothing phenomenon). *First, recall that $\mathbf{Z} = \sum_k \gamma_k \mathbf{H}^{(k)}$. If over-smoothing occurs in the GPR-GNN for $K$ sufficiently large, we have $\mathbf{Z}_{:j} = c_0 \boldsymbol{\beta}_j \boldsymbol{\pi}, \; \forall j \in [C]$ for some $c_0 > 0$ if $\gamma_k > 0$ and $\mathbf{Z}_{:j} = -c_0 \boldsymbol{\beta}_j \boldsymbol{\pi}, \; \forall j \in [C]$ for some $c_0 > 0$ if $\gamma_k < 0$.*

**Lemma A.4.** *Let $L = \sum_{i \in \mathcal{T}} L_i = \sum_{i \in \mathcal{T}} -log(\langle \hat{\mathbf{P}}_{i:}, \mathbf{Y}_{i:} \rangle)$ be the cross entropy loss and let $\mathcal{T}$ be the training set. Under the same assumption as given in Lemma A.2, the gradient of $\gamma_k$ for $k$ large enough is $\frac{\partial L}{\partial \gamma_k} = \sum_{i \in \mathcal{T}} \eta \boldsymbol{\pi}_i \langle \hat{\mathbf{P}}_{i:} - \mathbf{Y}_{i:}, \boldsymbol{\beta} \rangle + o_k(1)$.*

**Lemma A.5.** *For any real vector $\boldsymbol{\beta} \in \mathbb{R}^C$ and $\eta > 0$ large enough, we have softmax$_\eta(\boldsymbol{\beta}) = \mathbf{1}[\boldsymbol{\beta}] + o_\eta(1)$.*

Now we are ready to state the formal version of Theorem 4.2.

**Theorem A.6** (Formal version of Theorem 4.2). *Under the same assumptions as those listed in Lemma A.2, if the training set contains nodes from each class, then the GPR-GNN method can always avoid over-smoothing. More specifically, for $k, \eta$ large enough we have*

$$\frac{\partial L}{\partial \gamma_k} = \sum_{i \in \mathcal{T}} \eta \boldsymbol{\pi}_i \left( \max_{j \in [C]} \boldsymbol{\beta}_j - \boldsymbol{\beta}_{\mathbf{1}[\mathbf{Y}_{i:}]} \right) + o_k(1) + o_\eta(1), \; \text{when } \gamma_k > 0. \tag{3}$$

$$\frac{\partial L}{\partial \gamma_k} = \sum_{i \in \mathcal{T}} \eta \boldsymbol{\pi}_i \left( \min_{j \in [C]} \boldsymbol{\beta}_j - \boldsymbol{\beta}_{\mathbf{1}[\mathbf{Y}_{i:}]} \right) + o_k(1) + o_\eta(1), \; \text{when } \gamma_k < 0. \tag{4}$$

Note that when $\gamma_k > 0$, (3) $\geq 0$ when ignoring the $o(1)$ term. The equality is achieved if and only if $\max_{j \in [C]} \boldsymbol{\beta}_j = \boldsymbol{\beta}_{\mathbf{1}[\mathbf{Y}_{i:}]}$. This means that over-smoothing results in a prediction that perfectly aligns with the ground truth label in the training set. However, if our training set contains at least one node from each class then the equality can never be attained. Thus, the gradient of $\gamma_k$ will always be positive when $\gamma_k > 0$. Similarly when $\gamma_k < 0$, (4) $\leq 0$ when ignoring the $o(1)$ term. The equality is achieved if and only if $\min_{j \in [C]} \boldsymbol{\beta}_j = \boldsymbol{\beta}_{\mathbf{1}[\mathbf{Y}_{i:}]}$. By the same reason we know that under the assumption on training set the equality can never be attained. Thus, the gradient of $\gamma_k$ will always

be negative when $\gamma_k < 0$. Finally, it is not hard to check that the gradient is bounded in magnitude. Together we have shown that the gradient of $\gamma_k$ and $\gamma_k$ are of the same sign. This directly implies that $|\gamma_k|$ will approach to $0$ until we escape from over-smoothing when we use a decreasing learning rate for the optimizer (i.e. SGD).

*Proof.* First, let us assume the over-smoothing takes place and the $\gamma_k > 0$ for the dominate term. By Definition A.3, we know that $\mathbf{Z}_{:j} = c_0 \boldsymbol{\beta}_j \boldsymbol{\pi}$, $\forall j \in [C]$ for some $c_0 > 0$ and $K$ sufficiently large. By Lemma A.4 we have

$$\frac{\partial L}{\partial \gamma_k} = \sum_{i \in \mathcal{T}} \eta \boldsymbol{\pi}_i \left\langle \frac{e^{\eta \mathbf{Z}_{i:}}}{\sum_{j \in [C]} e^{\eta \mathbf{Z}_{ij}}} - \mathbf{Y}_{i:}, \boldsymbol{\beta} \right\rangle + o_k(1) \tag{5}$$

$$= \sum_{i \in \mathcal{T}} \eta \boldsymbol{\pi}_i \left\langle \frac{e^{\eta c_0 \boldsymbol{\pi}_i \boldsymbol{\beta}}}{\sum_{j \in [C]} e^{\eta c_0 \boldsymbol{\pi}_i \boldsymbol{\beta}_j}} - \mathbf{Y}_{i:}, \boldsymbol{\beta} \right\rangle + o_k(1), \tag{6}$$

where the last step follows from Definition A.3. Next, by Lemma A.5, we may approximate the softmax$_\eta$ by the true argmax for $\eta > 0$ large enough according to

$$\sum_{i \in \mathcal{T}} \eta \boldsymbol{\pi}_i \left\langle \mathbf{1}[c_0 \boldsymbol{\pi}_i \boldsymbol{\beta}] - \mathbf{Y}_{i:}, \boldsymbol{\beta} \right\rangle + o_k(1) + o_\eta(1) \tag{7}$$

$$= \sum_{i \in \mathcal{T}} \eta \boldsymbol{\pi}_i \left\langle \mathbf{1}[\boldsymbol{\beta}] - \mathbf{Y}_{i:}, \boldsymbol{\beta} \right\rangle + o_k(1) + o_\eta(1) \tag{8}$$

$$= \sum_{i \in \mathcal{T}} \eta \boldsymbol{\pi}_i \left( \max_{j \in [C]} \boldsymbol{\beta}_j - \boldsymbol{\beta}_{\mathbf{1}[\mathbf{Y}_{i:}]} \right) + o_k(1) + o_\eta(1). \tag{9}$$

The first equality is due to the fact that $c_0 > 0$ and $\boldsymbol{\pi}_i > 0$. Recall that by Lemma A.2, $\boldsymbol{\pi}_i = \frac{\sqrt{\tilde{\mathbf{D}}_{ii}}}{\sqrt{\sum_{v \in V} \tilde{\mathbf{D}}_{vv}}}$. Since we have a self- loop for each node, $\tilde{\mathbf{D}}_{ii} > 0$ and thus $\boldsymbol{\pi}_i > 0$. For the case $\gamma_k < 0$, the same analysis still valid until (7). Hence we have

$$\sum_{i \in \mathcal{T}} \eta \boldsymbol{\pi}_i \left\langle \mathbf{1}[-c_0 \boldsymbol{\pi}_i \boldsymbol{\beta}] - \mathbf{Y}_{i:}, \boldsymbol{\beta} \right\rangle + o_k(1) + o_\eta(1) \tag{10}$$

$$= \sum_{i \in \mathcal{T}} \eta \boldsymbol{\pi}_i \left\langle \mathbf{1}[-\boldsymbol{\beta}] - \mathbf{Y}_{i:}, \boldsymbol{\beta} \right\rangle + o_k(1) + o_\eta(1) \tag{11}$$

$$= \sum_{i \in \mathcal{T}} \eta \boldsymbol{\pi}_i \left( \min_{j \in [C]} \boldsymbol{\beta}_j - \boldsymbol{\beta}_{\mathbf{1}[\mathbf{Y}_{i:}]} \right) + o_k(1) + o_\eta(1). \tag{12}$$

Together we complete the proof.

$\square$

### A.5 CSBM DETAILS

The cSBM adds Gaussian random vectors as node features on top of the classical SBM. For simplicity, we assume $C = 2$ equally sized communities with node labels $v_i$ in $\{+1, -1\}$. Each node $i$ is associate with a $f$ dimensional Gaussian vector $b_i = \sqrt{\frac{\mu}{n}} v_i u + \frac{Z_i}{\sqrt{f}}$ where $n$ is the number of nodes, $u \sim N(0, I/f)$ and $Z_i \in \mathbf{R}^f$ has independent standard normal entries. The (undirected) graph in cSBM is described by the adjacency matrix $\mathbf{A}$ defined as

$$\mathbf{P}(\mathbf{A}_{ij} = 1) = \begin{cases} \frac{d + \lambda \sqrt{d}}{n} & \text{if } v_i v_j > 0 \\ \frac{d - \lambda \sqrt{d}}{n} & \text{otherwise} \end{cases}.$$

Similar to the classical SBM, given the node labels the edges are independent. The symbol $d$ stands for the average degree of the graph. Also, recall that $\mu$ and $\lambda$ control the information strength carried by the node features and the graph structure respectively.

One reason for using the cSBM to generate synthetic data is that the information-theoretic limit of the model is already characterized in Deshpande et al. (2018). This result is summarized below.

**Theorem A.7** (Informal main result in Deshpande et al. (2018)). *Assume that $n, f \to \infty$, $\frac{n}{f} \to \xi$ and $d \to \infty$. Then there exists an estimator $\hat{v}$ such that $\liminf_{n \to \infty} \frac{|\langle \hat{v}, v \rangle|}{n}$ is bounded away from $0$ if and only if $\lambda^2 + \frac{\mu^2}{\xi} > 1$.*

In our experiment, we set $n = 5000$, $f = 2000$ and thus have $\xi = 2.5$. We vary $\mu$ and $\lambda$ along the arc $\lambda^2 + \mu^2/\xi = 1 + \epsilon$ for some $\epsilon > 0$ to ensure that we are in the achievable parameter regime. We also choose $\epsilon = 3.25$ for all our experiment.

### A.6 Proof of Lemma A.2

Note that the proof of Lemma A.2 reduces to a standard analysis of random walks on graph. We include it for completeness and refer the interested readers to the tutorial Von Luxburg (2007).

We start by showing that the symmetric graph Laplacian

$$\tilde{\mathbf{L}}_{\text{sym}} = \mathbf{I} - \tilde{\mathbf{D}}^{-1/2}\tilde{\mathbf{A}}\tilde{\mathbf{D}}^{-1/2} = \mathbf{I} - \tilde{\mathbf{A}}_{\text{sym}} \tag{13}$$

is positive semi-definite. Let $u$ be any real vector of unit norm and $f = \tilde{\mathbf{D}}^{-1/2}u$, then we have

$$u^T\tilde{\mathbf{L}}_{\text{sym}}u = u^Tu - u^T\tilde{\mathbf{D}}^{-1/2}\tilde{\mathbf{A}}\tilde{\mathbf{D}}^{-1/2}u = \sum_{i=1}^{n} u_i^2 - \sum_{i,j=1}^{n} f_i f_j \tilde{\mathbf{A}}_{ij} \tag{14}$$

$$= \sum_{i=1}^{n} \tilde{\mathbf{D}}_{ii} f_i^2 - \sum_{i,j=1}^{n} f_i f_j \tilde{\mathbf{A}}_{ij} = \frac{1}{2}\left( \sum_{i=1}^{n} \tilde{\mathbf{D}}_{ii} f_i^2 - 2\sum_{i,j=1}^{n} f_i f_j \tilde{\mathbf{A}}_{ij} + \sum_{j=1}^{n} \tilde{\mathbf{D}}_{jj} f_j^2 \right) \tag{15}$$

$$= \frac{1}{2}\sum_{i,j=1}^{n} \tilde{\mathbf{A}}_{ij}(f_i - f_j)^2, \tag{16}$$

where the last step follows from the definition of the degree.

Next we show that $0$ is indeed an eigenvalue of $\tilde{\mathbf{L}}_{\text{sym}}$ associated with the unit eigenvector $\boldsymbol{\pi}$ where $\boldsymbol{\pi} = \frac{\sqrt{\tilde{\mathbf{D}}_{ii}}}{\sqrt{\sum_v \tilde{\mathbf{D}}_{vv}}}$.

Let $\mathbb{1}$ be the all one vector. Then, a direct calculation reveals that

$$\tilde{\mathbf{L}}_{\text{sym}}\boldsymbol{\pi} = \boldsymbol{\pi} - \tilde{\mathbf{D}}^{-1/2}\tilde{\mathbf{A}}\tilde{\mathbf{D}}^{-1/2}\boldsymbol{\pi} = \boldsymbol{\pi} - \tilde{\mathbf{D}}^{-1/2}\tilde{\mathbf{A}}\tilde{\mathbf{D}}^{-1/2}\tilde{\mathbf{D}}^{1/2}\mathbb{1} \times \frac{1}{\sqrt{\sum_v \tilde{\mathbf{D}}_{vv}}} \tag{17}$$

$$= \boldsymbol{\pi} - \tilde{\mathbf{D}}^{-1/2}\tilde{\mathbf{A}}\mathbb{1} \times \frac{1}{\sqrt{\sum_v \tilde{\mathbf{D}}_{vv}}} = \boldsymbol{\pi} - \tilde{\mathbf{D}}^{-1/2}\tilde{\mathbf{D}}\mathbb{1} \times \frac{1}{\sqrt{\sum_v \tilde{\mathbf{D}}_{vv}}} \tag{18}$$

$$= \boldsymbol{\pi} - \tilde{\mathbf{D}}^{1/2}\mathbb{1} \times \frac{1}{\sqrt{\sum_v \tilde{\mathbf{D}}_{vv}}} = \boldsymbol{\pi} - \boldsymbol{\pi} = 0. \tag{19}$$

Combining this result with the positive semi-definite property of the Laplacian shows that $0$ is indeed the smallest eigenvalue of $\tilde{\mathbf{L}}_{\text{sym}}$ associated with the eigenvector $\boldsymbol{\pi}$. Moreover, from (16) and the assumption that the graph is connected, it is not hard to see that the multiplicity of the eigenvalue $0$ is exactly $1$ (See Proposition 2 and 4 in Von Luxburg (2007) for more detail). Finally, from (13) it is obvious that the the largest eigenvalue of $\tilde{\mathbf{A}}_{\text{sym}}$ is $1$, which correspond to the eigenvector $\boldsymbol{\pi}$. Hence all other eigenvalues of $\tilde{\mathbf{A}}_{\text{sym}}$ $1 > \lambda_2 \geq ... \geq \lambda_n$.

Next, we prove that $|\lambda_n| < 1$. This can also be shown directly from (16). Note that

$$u^T \tilde{\mathbf{L}}_{\text{sym}} u = \frac{1}{2} \sum_{i,j=1}^{n} \tilde{\mathbf{A}}_{ij} (f_i - f_j)^2 \tag{20}$$

$$\leq \sum_{i,j=1}^{n} \tilde{\mathbf{A}}_{ij} (f_i^2 + f_j^2) = 2 \sum_{i,j=1}^{n} \tilde{\mathbf{A}}_{ij} f_i^2 = 2 \sum_{i,j=1}^{n} \tilde{\mathbf{A}}_{ij} \frac{u_i^2}{\tilde{\mathbf{D}}_{ii}} \tag{21}$$

$$= 2 \sum_{i=1}^{n} \frac{u_i^2}{\tilde{\mathbf{D}}_{ii}} \sum_{j=1}^{n} \tilde{\mathbf{A}}_{ij} = 2 \sum_{i=1}^{n} \frac{u_i^2}{\tilde{\mathbf{D}}_{ii}} \tilde{\mathbf{D}}_{ii} = 2 \sum_{i=1}^{n} u_i^2 = 2. \tag{22}$$

The inequality follows from an application of the Cauchy-Schwartz inequality. Consequently, the largest eigenvalue of $\tilde{\mathbf{L}}_{\text{sym}}$ is bounded by 2 which means that $|\lambda_n| \leq 1$. Note that equality holds if and only if the underlying graph is bipartite. However, this is impossible in our setting since we have added a self loop to each node. Hence $|\lambda_n| < 1$. This means

$$\lim_{k \to \infty} \tilde{\mathbf{A}}_{\text{sym}}^k = \boldsymbol{\pi} \boldsymbol{\pi}^T. \tag{23}$$

Hence, for any $\mathbf{H}^{(0)}$ we have

$$\lim_{k \to \infty} \tilde{\mathbf{A}}_{\text{sym}}^k \mathbf{H}^{(0)} = \boldsymbol{\pi} \boldsymbol{\pi}^T \mathbf{H}^{(0)} = \boldsymbol{\pi} \boldsymbol{\beta}^T. \tag{24}$$

Note that this can also be written with the $o_k(1)$ term as

$$\tilde{\mathbf{A}}_{\text{sym}}^k \mathbf{H}^{(0)} = \boldsymbol{\pi} \boldsymbol{\beta}^T + o_k(1). \tag{25}$$

This completes the proof.

## A.7 PROOF OF LEMMA A.4

Recall that our loss function equals

$$L = \sum_{i \in \mathcal{T}} L_i = \sum_{i \in \mathcal{T}} - \log\left(\frac{e^{\eta \langle \mathbf{Z}_{i:}, \mathbf{Y}_{i:} \rangle}}{\sum_{m=1}^{C} e^{\eta \mathbf{Z}_{im}}}\right). \tag{26}$$

Then by taking the partial derivative of the loss function with respect to $\gamma_{k'}$ we have

$$\frac{\partial L}{\partial \gamma_{k'}} = \frac{\partial}{\partial \gamma_{k'}} \sum_{i \in \mathcal{T}} \left(\log\left(\sum_{m=1}^{C} e^{\eta \mathbf{Z}_{im}}\right) - \langle \eta \mathbf{Z}_{i:}, \mathbf{Y}_{i:} \rangle\right). \tag{27}$$

Next, recall that for GPR-GNN we also have $\mathbf{Z} = \sum_{k=0}^{K} \gamma_k \mathbf{H}^{(k)}$. Plugging this expression into the previous formula and applying the chain rule we obtain

$$\frac{\partial}{\partial \gamma_{k'}} \sum_{i \in \mathcal{T}} \left(\log\left(\sum_{m=1}^{C} e^{\eta \mathbf{Z}_{im}}\right) - \langle \eta \mathbf{Z}_{i:}, \mathbf{Y}_{i:} \rangle\right) = \sum_{i \in \mathcal{T}} \left(\frac{\sum_{m=1}^{C} e^{\eta \mathbf{Z}_{im}} \frac{\partial \eta \mathbf{Z}_{im}}{\partial \gamma_{k'}}}{\sum_{m=1}^{C} e^{\mathbf{Z}_{im}}} - \left\langle \eta \mathbf{H}_{i:}^{(k')}, \mathbf{Y}_{i:} \right\rangle\right) \tag{28}$$

$$= \sum_{i \in \mathcal{T}} \left(\frac{\sum_{m=1}^{C} e^{\eta \mathbf{Z}_{im}} \eta \mathbf{H}_{im}^{(k')}}{\sum_{m=1}^{C} e^{\eta \mathbf{Z}_{im}}} - \left\langle \eta \mathbf{H}_{i:}^{(k')}, \mathbf{Y}_{i:} \right\rangle\right) \tag{29}$$

Settin $k' = k$ for large enough $k$, it follows from Lemma A.2 that

$$\frac{\partial L}{\partial \gamma_k} = \sum_{i \in \mathcal{T}} \eta \left(\frac{\sum_{m=1}^{C} e^{\eta \mathbf{Z}_{im}} \mathbf{H}_{im}^{(k)}}{\sum_{m=1}^{C} e^{\eta \mathbf{Z}_{im}}} - \left\langle \mathbf{H}_{i:}^{(k)}, \mathbf{Y}_{i:} \right\rangle\right) \tag{30}$$

$$= \sum_{i \in \mathcal{T}} \eta \left(\frac{\sum_{m=1}^{C} e^{\eta \mathbf{Z}_{im}} (\boldsymbol{\pi}_i \boldsymbol{\beta}_m + o_k(1))}{\sum_{m=1}^{C} e^{\eta \mathbf{Z}_{im}}} - \langle \boldsymbol{\pi}_i \boldsymbol{\beta} + o_k(1), \mathbf{Y}_{i:} \rangle\right) \tag{31}$$

$$= \sum_{i \in \mathcal{T}} \boldsymbol{\pi}_i \eta \left(\frac{\sum_{m=1}^{C} e^{\eta \mathbf{Z}_{im}} \boldsymbol{\beta}_m}{\sum_{m=1}^{C} e^{\eta \mathbf{Z}_{im}}} - \langle \boldsymbol{\beta}, \mathbf{Y}_{i:} \rangle\right) + o_k(1) \tag{32}$$

$$= \sum_{i \in \mathcal{T}} \boldsymbol{\pi}_i \eta \left(\sum_{m=1}^{C} \hat{\mathbf{P}}_{im} \boldsymbol{\beta}_m - \langle \boldsymbol{\beta}, \mathbf{Y}_{i:} \rangle\right) + o_k(1) = \sum_{i \in \mathcal{T}} \eta \boldsymbol{\pi}_i \left\langle \hat{\mathbf{P}}_{i:} - \mathbf{Y}_{i:}, \boldsymbol{\beta} \right\rangle + o_k(1). \tag{33}$$

Note that in (32) and (33) we used the definition of the soft prediction $\hat{\mathbf{P}} = \text{softmax}_\eta(\mathbf{Z})$. This completes the proof.

## A.8 PROOF OF LEMMA A.5

Let $\hat{\beta} = \max(\boldsymbol{\beta})$. Then by the definition of $\text{softmax}_\eta$ for $\eta > 0$ we have

$$\text{softmax}_\eta(\boldsymbol{\beta}) = \frac{e^{\eta\boldsymbol{\beta}}}{\sum_{m=1}^C e^{\eta\boldsymbol{\beta}_m}} = \frac{e^{-\eta(\hat{\beta}-\boldsymbol{\beta})}}{\sum_{m=1}^C e^{-\eta(\hat{\beta}-\boldsymbol{\beta}_m)}}. \tag{34}$$

Note that $\hat{\beta} - \boldsymbol{\beta}_m > 0$ when $\boldsymbol{\beta}_m \neq \hat{\beta}$ and $\hat{\beta} - \boldsymbol{\beta}_m = 0$ when $\boldsymbol{\beta}_m = \hat{\beta}$. Without loss of generality we assume that there are $p$ maxima in $\boldsymbol{\beta}$, where $1 \leq p \leq C$, and let $\mathcal{P}$ denote the set of indices of those maxima. Then, taking the limit $\eta \to \infty$ we have

$$\lim_{\eta\to\infty} \text{softmax}_\eta(\boldsymbol{\beta})_j = \lim_{\eta\to\infty} \frac{e^{-\eta(\hat{\beta}-\boldsymbol{\beta}_j)}}{\sum_{m\notin\mathcal{P}} e^{-\eta(\hat{\beta}-\boldsymbol{\beta}_m)} + p} = \begin{cases} 0, & \text{if } \boldsymbol{\beta}_j \neq \hat{\beta} \\ \frac{1}{p}, & \text{otherwise.} \end{cases} \tag{35}$$

This implies that for $\eta > 0$ large enough one has

$$\text{softmax}_\eta(\boldsymbol{\beta}) = \mathbf{1}[\boldsymbol{\beta}] + o_\eta(1). \tag{36}$$

The above result completes the proof.

## A.9 ADDITIONAL EXPERIMENTAL DETAILS

Table 4: The values of the homophily measure for cSBM datasets.

| $\phi$ | $-1$ | $-0.75$ | $-0.5$ | $-0.25$ | $0$ | $0.25$ | $0.5$ | $0.75$ | $1$ |
|---|---|---|---|---|---|---|---|---|---|
| $\mathcal{H}(G)$ | 0.001 | 0.002 | 0.009 | 0.029 | 0.077 | 0.189 | 0.419 | 0.688 | 0.809 |

All experiments are performed on a Linux Machine with 48 cores, 376GB of RAM, and a NVIDIA Tesla P100 GPU with 12GB of GPU memory. For the training set, we ensure that number of nodes from each class is approximately the same an keep the total number of training nodes close to $2.5\%/60\%$. For the validation set, we randomly sample $2.5\%/20\%$ of the nodes and place the remaining ones into the test set.

For all baseline models, we directly use the implementation available in the Pytorch Geometric library Fey & Lenssen (2019).We use early stopping 200 and a maximum number of epochs equal to 1000 for both real benchmark dataset and our cSBM synthetic datasets. All models use the Adam optimizer Kingma & Ba (2014). Note that the early stopping criteria is exactly the same as in Pytorch Geometric – when the epoch is greater than half of the maximum epoch, we check if the current validation loss is lower than the average over the past 200 epochs. If it is not lower, we stop the training process.

For GCN, we use 2 GCN layers with 64 hidden units. For GAT, we use 2 GAT convolutional layers, where the first layer has 8 attention heads and each head has 8 hidden units; the second layer has 1 attention head and 64 hidden units. For GCN-Cheby, we use 2 steps propagation for each layer with 32 hidden units. Note that the number of equivalent hidden units for each layer is64 for this case. For JK-Net, we use the GCN-based model with 2 layers and 16 hidden units in each layer. As for the layer aggregation part, we use a LSTM with 16 channels and 4 layers. For the MLP, we choose a 2-layer fully connected network with 64 hidden units. For APPNP we use the same 2-layer MLP with 10 steps of propagation. Besides the GPR-GNN, we fix the dropout rate for the NN part to be 0.5 as APPNP and optimize the dropout rate for the GPR part among $\{0, 0.5, 0.7\}$. For Geom-GCN, we choose the datasets already tested in the paper were the method was first described (Pei et al., 2019). For SGC, we use the default $K = 2$ layers after test among $\{2, 3\}$. For SAGE, we use 2 SAGE convolutional layers with 64 hidden units.

**The heterophilic datasets used in (Pei et al., 2019).** The graphs Chameleon, Actor, Squirrel, Texas and Cornell in their original form are directed graphs (see the github repository of (Pei et al., 2019)). Since the usual setting for semi-supervised node classifications involves undirected graph, we transformed the graphs into undirected to test them on all previously described benchmark methods. We keep the input graph directed for Geom-GCN as the method uses a fixed preprocessing scheme that

was unfortunately not made public by the authors. Our homophily measure values $\mathcal{H}(G)$ in Table 1 are all based on undirected graphs and hence the numbers are different from those reported in (Pei et al., 2019).

## A.10 ADDITIONAL EXPERIMENTAL RESULTS

Table 5: Results for cSBM, sparse splitting. Bold values indicate the best obtained result and while bold, underlined values indicate results within a 95% confidence interval with respect to the best result.

|  | $\phi = -1$ | $\phi = -0.75$ | $\phi = -0.5$ | $\phi = -0.25$ | $\phi = 0$ | $\phi = 0.25$ | $\phi = 0.5$ | $\phi = 0.75$ | $\phi = 1$ |
|---|---|---|---|---|---|---|---|---|---|
| GPRGNN | **97.19**±**0.16** | **95.54**±**0.15** | 81.54±0.73 | 60.65±0.31 | **62.16**±**0.23** | 68.83±0.28 | 89.31±0.16 | **96.98**±**0.08** | **96.71**±**0.13** |
| GPRGNN(random) | 88.39±3.31 | 88.54±3.01 | 66.91±2.93 | 56.35±0.98 | 58.09±0.71 | 64.01±1.39 | 81.93±1.68 | 94.59±0.29 | 93.69±1.04 |
| APPNP | 49.57±0.11 | 52.45±0.27 | 56.32±0.40 | 59.55±0.48 | 61.21±0.23 | 68.41±0.30 | 85.66±0.22 | 94.37±0.09 | 90.02±0.16 |
| MLP | 49.88±0.10 | 53.40±0.34 | 57.14±0.41 | 60.55±0.41 | **62.15**±**0.33** | 61.26±0.21 | 57.91±0.35 | 53.36±0.32 | 49.92±0.11 |
| SGC | 54.41±0.37 | 59.74±0.29 | 55.57±0.33 | 51.84±0.23 | 53.95±0.28 | 65.65±0.27 | 85.51±0.20 | 93.99±0.10 | 88.50±0.18 |
| GCN | 55.24±0.35 | 61.04±0.39 | 56.40±0.39 | 52.23±0.24 | 54.43±0.32 | 67.23±0.29 | 84.56±0.20 | 90.19±0.14 | 78.67±0.19 |
| GAT | 53.97±0.32 | 57.18±0.45 | 53.39±0.34 | 51.23±0.19 | 53.26±0.27 | 64.45±0.36 | 81.94±0.34 | 88.45±0.26 | 78.06±0.30 |
| SAGE | 62.30±0.50 | 75.10±0.50 | 72.84±0.44 | 58.62±0.30 | **63.88**±**0.37** | 63.55±0.47 | 73.50±0.50 | 75.26±0.52 | 62.61±0.44 |
| JKNet | 51.70±0.39 | 55.83±0.75 | 52.67±0.51 | 50.27±0.15 | 52.02±0.35 | 65.67±0.44 | 86.35±0.19 | 95.13±0.09 | 90.32±0.17 |
| GCN-Cheby | 61.44±0.51 | 73.91±0.75 | 71.96±0.6 | **63.96**±**0.43** | 59.70±0.34 | 64.00±0.38 | 72.34±0.63 | 73.56±0.65 | 60.88±0.58 |

Table 6: Results for cSBM, dense splitting. Bold values indicate the best results found while bold, underlined values indicate results within a 95% confidence interval with respect to the best result.

|  | $\phi = -1$ | $\phi = -0.75$ | $\phi = -0.5$ | $\phi = -0.25$ | $\phi = 0$ | $\phi = 0.25$ | $\phi = 0.5$ | $\phi = 0.75$ | $\phi = 1$ |
|---|---|---|---|---|---|---|---|---|---|
| GPRGNN | **98.83**±**0.06** | **98.19**±**0.08** | **94.23**±**0.14** | **86.06**±**0.20** | **82.22**±**0.20** | **86.48**±**0.20** | **94.34**±**0.13** | **98.46**±**0.08** | **98.84**±**0.06** |
| GPRGNN(random) | 98.75±0.05 | 98.08±0.08 | 94.22±0.14 | 86.06±0.20 | 81.57±0.23 | 86.36±0.20 | 94.09±0.14 | 98.38±0.08 | 98.77±0.07 |
| APPNP | 48.94±0.29 | 63.87±0.29 | 73.30±0.26 | 79.30±0.20 | 82.41±0.23 | 86.47±0.18 | 94.20±0.14 | 97.96±0.10 | 98.53±0.08 |
| MLP | 49.79±0.29 | 66.69±0.27 | 75.36±0.26 | 80.30±0.24 | 82.19±0.24 | 80.88±0.22 | 76.07±0.24 | 66.61±0.25 | 49.65±0.29 |
| SGC | 78.95±0.23 | 81.79±0.24 | 75.15±0.25 | 59.40±0.28 | 63.75±0.26 | 80.81±0.22 | 93.04±0.15 | 98.05±0.08 | 97.80±0.09 |
| GCN | 78.50±0.28 | 83.68±0.22 | 75.98±0.25 | 59.98±0.25 | 64.09±0.26 | 81.89±0.19 | 93.91±0.12 | 97.78±0.08 | 96.29±0.11 |
| GAT | 82.39±0.41 | 80.37±0.22 | 71.01±0.26 | 57.68±0.29 | 62.95±0.28 | 80.61±0.24 | 93.26±0.14 | 97.99±0.08 | 98.40±0.09 |
| SAGE | 91.33±0.23 | 95.72±0.12 | 93.23±0.17 | 84.52±0.20 | 78.99±0.24 | 84.87±0.20 | 92.90±0.15 | 95.75±0.11 | 91.19±0.24 |
| JKNet | 96.11±0.37 | 95.33±0.25 | 87.98±0.56 | 59.61±0.49 | 63.28±0.10 | 80.23±0.36 | 93.28±0.15 | 98.33±0.07 | 98.22±0.07 |
| GCN-Cheby | 90.94±0.16 | 94.82±0.13 | 91.83±0.17 | 85.18±0.21 | 80.80±0.25 | 85.28±0.21 | 92.70±0.16 | 95.06±0.13 | 90.34±0.18 |

Table 7: Results on homophilic real-world benchmark datasets tested in (Pei et al., 2019), dense splitting: Mean accuracy (%) ± 95% confidence interval. Boldface values indicate the best results found while boldface, underlined values indicates results within the confidence interval with respect to the best result.

|  | Cora | Citeseer | PubMed |
|---|---|---|---|
| GPRGNN | **88.65**±**0.28** | 80.01±0.28 | **89.18**±**0.15** |
| APPNP | 88.1±0.23 | **80.5**±**0.26** | **89.15**±**0.13** |
| MLP | 76.44±0.30 | 76.25±0.28 | 86.43±0.13 |
| SGC | 86.58±0.26 | 76.23±0.29 | 83.52±0.10 |
| GCN | 86.87±0.25 | 79.28±0.25 | 86.97±0.12 |
| GAT | 87.52±0.24 | **80.56**±**0.31** | 86.64±0.11 |
| SAGE | 86.58±0.26 | 78.24±0.30 | 86.85±0.11 |
| JKNet | 86.97±0.27 | 77.69±0.35 | 87.38±0.13 |
| GCN-Cheby | 86.46±0.26 | 78.66±0.26 | 88.2±0.09 |
| GeomGCN | 85.4±0.26 | 76.42±0.37 | 88.51±0.08 |

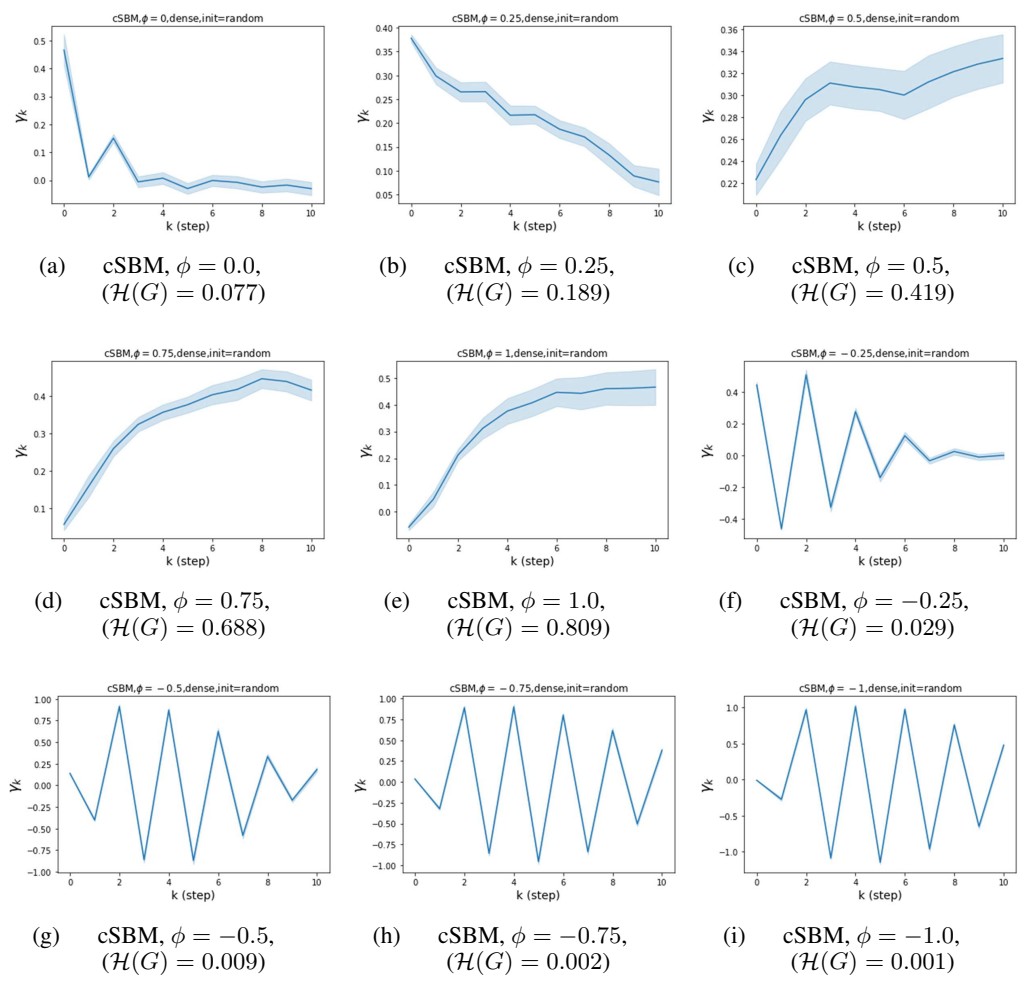

Figure 7: Figures (a)-(i) show the learned GPR weights by GPR-GNN with random initialization on cSBM, dense splitting. The shaded region indicates a $95\%$ confidence interval.

Table 8: Additional experiments illustrating that GPR-GNN escapes over-smoothing. We initialize the GPR weights $\gamma_k = \delta_{kK}$ as described in Section 5. We report the mean accuracy at Epoch 0 and after training (Final epoch). The over-smoothing ratio indicates how many time out of the 100 runs that GPR-GNN started with lead to the same label for all nodes. For an illustration of how GPR weights change over different epochs, please check Figure 9.

|  | Accuracy at epoch 0(%) | Accuracy at the final epoch(%) | Over-smoothing ratio(%) |
|---|---|---|---|
| Cora | 12.75 | 88.25 | 84 |
| Computers | 9.41 | 85.93 | 89 |
| Squirrel | 19.87 | 52.06 | 97 |
| Texas | 21.05 | 90.05 | 100 |

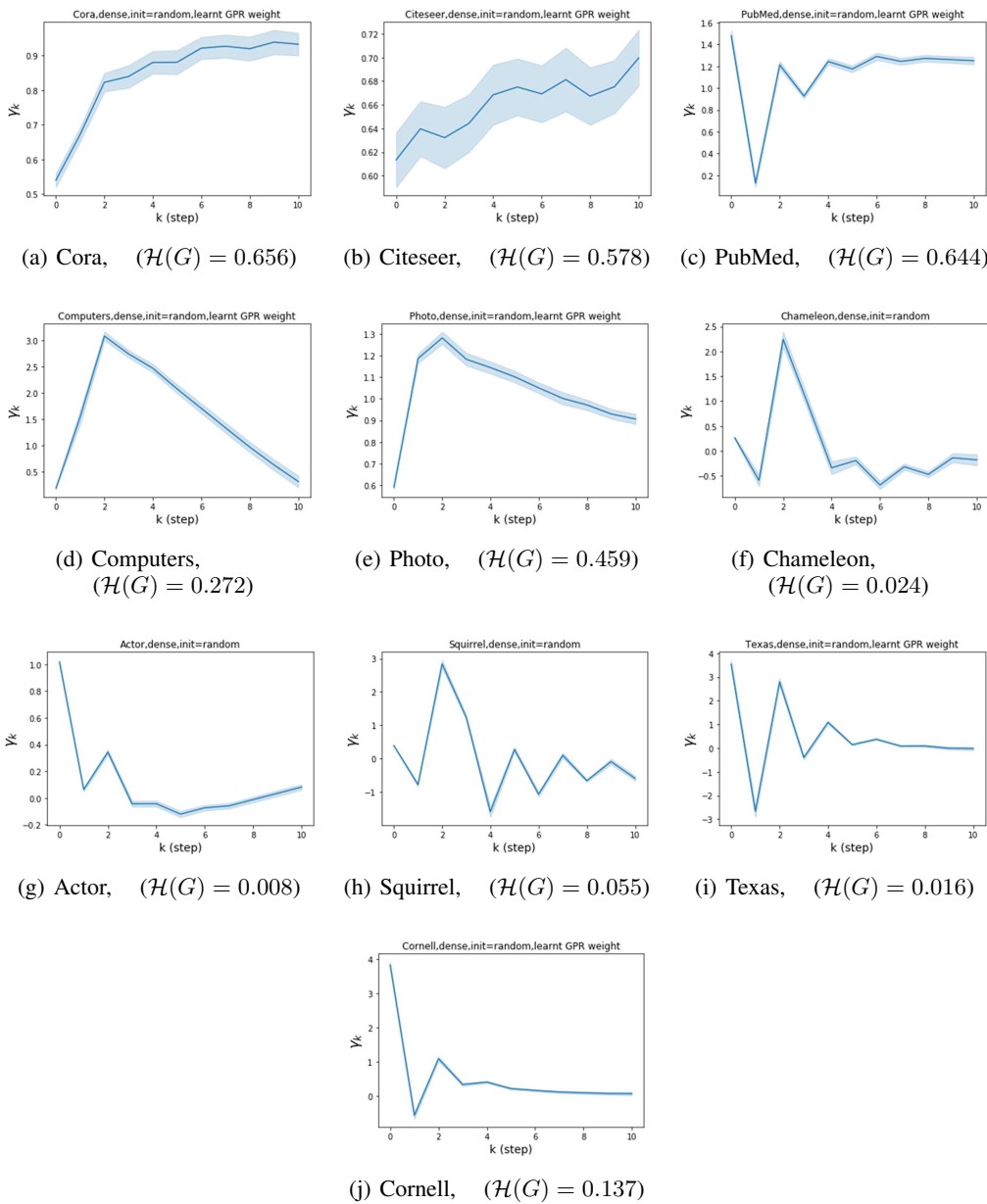

Figure 8: Figures (a)-(j) show the learned GPR weights by GPR-GNN with random initialization on various benchmark datasets, dense splitting. The shaded region indicates a $95\%$ confidence interval. Note that the learned GPR weights are all positive for every homophilic dataset. There is at least one negative learned GPR weight for every heterophilic dataset.

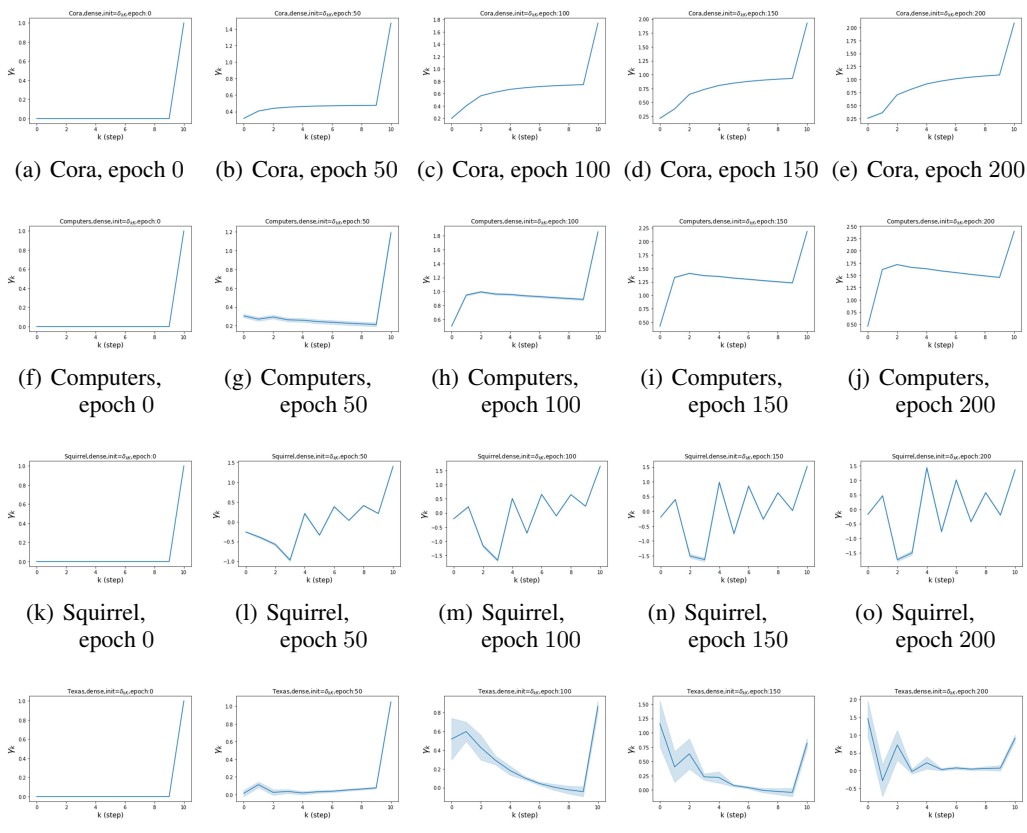

Figure 9: Learned GPR weights by GPR-GNN with initialization $\gamma_k = \delta_{kK}$ (last step) on various benchmark datasets, dense splitting. The shaded region indicates a 95% confidence interval. Also, please check Table 8. Note that the GPR weights $\{\gamma_k\}_{k=0}^{K}$ are identical to $\{-\gamma_k\}_{k=0}^{K}$ in terms of graph filtering.

Table 9: Efficiency on homophilic real world benchmark datasets: Average running time per epoch(ms)/average total running time(s).

|  | Cora | Citeseer | Pubmed | Computers | Photo |
|---|---|---|---|---|---|
| GPRGNN | 17.62ms / 3.74s | 19.28ms / 3.89s | 20.19ms / 5.53s | 39.93ms / 11.40s | 21.61ms / 6.18s |
| APPNP | 17.16ms / 4.00s | 15.97ms / 3.26s | 18.47ms / 6.29s | 39.59ms / 20.00s | 20.10ms / 10.93s |
| MLP | 4.14ms / 0.92s | 5.30ms / 1.13s | 5.43ms / 2.86s | 5.33ms / 2.77s | 4.63ms / 2.72s |
| SGC | 3.31ms / 3.31s | 11.45ms / 2.31s | 3.81ms / 3.81s | 4.36ms / 4.36s | 19.12ms / 8.75s |
| GCN | 9.25ms / 1.97s | 17.46ms / 3.53s | 14.11ms / 4.17s | 32.45ms / 16.29s | 32.56ms / 11.33s |
| GAT | 14.78ms / 3.42s | 19.94ms / 4.47s | 21.52ms / 6.70s | 61.45ms / 24.28s | 24.57ms / 11.61s |
| SAGE | 12.06ms / 2.44s | 41.40ms / 8.36s | 28.82ms / 6.32s | 171.36ms / 71.94s | 108.88ms / 42.18s |
| JKNet | 18.97ms / 4.41s | 3.99ms / 3.99s | 24.48ms / 6.61s | 35.02ms / 14.96s | 3.66ms / 3.66s |
| GCN-cheby | 22.96ms / 4.75s | 23.16ms / 4.68s | 45.76ms / 12.02s | 218.82ms / 96.58s | 82.38ms / 30.48s |

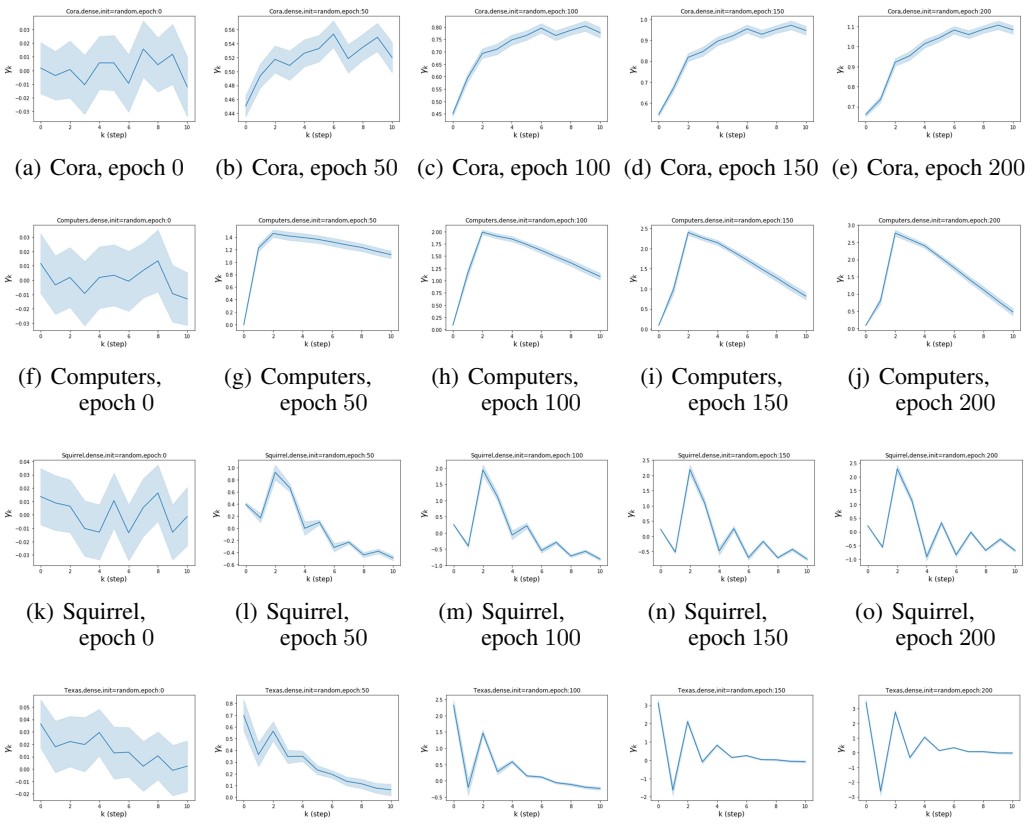

(a) Cora, epoch 0   (b) Cora, epoch 50   (c) Cora, epoch 100   (d) Cora, epoch 150   (e) Cora, epoch 200

(f) Computers, epoch 0   (g) Computers, epoch 50   (h) Computers, epoch 100   (i) Computers, epoch 150   (j) Computers, epoch 200

(k) Squirrel, epoch 0   (l) Squirrel, epoch 50   (m) Squirrel, epoch 100   (n) Squirrel, epoch 150   (o) Squirrel, epoch 200

(p) Texas, epoch 0   (q) Texas, epoch 50   (r) Texas, epoch 100   (s) Texas, epoch 150   (t) Texas, epoch 200

Figure 10: The dynamics of learning GPR weights with random initialization on various benchmark datasets, dense splitting. The shaded region indicates a 95% confidence interval.

Table 10: Efficiency on heterophilic real world benchmark datasets: Average running time per epoch(ms)/average total running time(s).

|  | Chameleon | Squirrel | Actor | Texas | Cornell |
|---|---|---|---|---|---|
| GPRGNN | 16.74ms / 3.40s | 25.28ms / 5.12s | 19.31ms / 4.49s | 17.56ms / 3.55s | 18.42ms / 3.72s |
| APPNP | 17.01ms / 3.44s | 22.93ms / 4.63s | 16.32ms / 4.04s | 15.96ms / 3.24s | 14.66ms / 3.09s |
| MLP | 3.41ms / 0.69s | 5.19ms / 1.05s | 4.84ms / 0.98s | 3.81ms / 1.04s | 3.46ms / 0.89s |
| SGC | 13.83ms / 2.79s | 27.11ms / 5.56s | 12.39ms / 2.50s | 10.22ms / 2.06s | 10.38ms / 2.10s |
| GCN | 16.63ms / 3.63s | 47.46ms / 10.05s | 18.91ms / 3.86s | 15.50ms / 3.13s | 13.67ms / 2.76s |
| GAT | 20.03ms / 5.15s | 29.89ms / 6.67s | 23.52ms / 4.75s | 19.67ms / 4.01s | 19.35ms / 3.91s |
| SAGE | 89.41ms / 18.06s | 440.55ms / 88.99s | 43.94ms / 8.88s | 12.34ms / 3.08s | 12.15ms / 2.69s |
| JKNet | 3.13ms / 3.13s | 4.79ms / 4.79s | 3.98ms / 1.00s | 2.86ms / 2.09s | 2.81ms / 1.18s |
| GCN-cheby | 64.43ms / 13.02s | 343.47ms / 69.38s | 27.95ms / 5.65s | 6.08ms / 1.28s | 6.05ms / 1.44s |

