# OpenReview forum: "Adaptive Universal Generalized PageRank Graph Neural Network"
_ICLR.cc/2021/Conference — ICLR 2021 Poster_

### Official Review · AnonReviewer2 · 2020-10-25
**Official Blind Review #2**

**Rating:** 6
**Confidence:** 2

**Review:**

In this paper, the authors proposed a generalized pagerank version of a graph neural network (GNN). The authors learn a weighted combination of higher powers of the graph adjacency matrix, with the weights themselves being learnable. This allows their method to generalize existing GNN methods that work well when there's graph homophily, but not in the case of heterophily. The proposed method reduces to existing cases under certain weight settings, but in other cases, the learned weights allow for the GNN to act as a "high pass filter", which existing methods do not do. By learning these weights, the authors can also go deeper in the graph, and aggregate information from several hops away. Experiments on several standard datasets show that the proposed method outperforms multiple baselines.

The paper is clearly written. My main concern is the overall novelty of the paper, with respect to ICLR. Unless I'm missing something, the main point the authors are making is to learn the weights of the decomposition, and show that by making those weights learnable, good things happen when compared to the methods in Klicpera et al, and Wu et al.  Just that to me is not grounds for strong acceptance.

minor comments:
page 2: you keep using the term "large step propagation" several times, without actually defining what that means.

Thm 4.1: is there a formal version of this theorem with a proof? if not it's probably not fair to call it a theorem.

please make the text in figures 3 and 4 larger. It's hard to read.

---

> ### Author Response · Authors · 2020-11-14
> **Some clarifications**
>
> We thank Reviewer 2 for her/his valuable feedback and recommendations for improving the manuscript. We address all the concerns raised below.
>
> 1.	"My main concern is the overall novelty of the paper, with respect to ICLR. Unless I'm missing something, the main point the authors are making is to learn the weights of the decomposition, and show that by making those weights learnable, good things happen when compared to the methods in Klicpera et al, and Wu et al. Just that to me is not grounds for strong acceptance."
>
> We would like to start by pointing out that the assessment made by the reviewer is in stark contrast with comments made by other reviewers, say Reviewer 1 who stated "The paper highlights and clearly explains why optimizing the proposed layerwise GPR weights in GNN could tackle the two challenges. Although the final GPR-GNN architecture is simple, the ideas behind are significant." Reviewer 3 also praised the novelty of our work. Note that the apparent simplicity of the GPR-GNN architecture does not imply that it is trivial or not worth publishing, and we strongly believe that simple solutions that outperform complicated methods are the most desirable ones in a practically-driven field such as machine learning. Furthermore, our theoretical analysis shows why using learnable GPR weights can simultaneously help us with resolving over-smoothing problems and dealing with both homophilic and heterophilic graph data.
>
> As a further note, the argument used to show that APPNP prevents over-smoothing does not hold for GPR-GNN. The former argument only holds for "fixed" GPR weights (PPR weights in APPNP). In contrast, we prove that allowing $\gamma_k$ to be learnable can also prevent over-smoothing (Theorem 4.2). Moreover, the way in which GPR-GNN prevents over-smoothing may be easier to understand and more effective (see our detailed discussion and a concrete illustrative example in Section A.2, Supplement). On the other hand, the proposed GPR technique also allows us to nicely connect our method with polynomial graph filtering and analyze the properties of the underlying filter (Theorem 4.1). All these theoretical results are nontrivial. Hence, the novelty of our work does not only lie in proposing a new architecture, but also in terms of deepening our (theoretical) understandings of the operation of GNNs. Finally, our work is the first proposals for using cSBM as synthetic dataset to evaluate GNNs. As we mentioned in our paper, cSBM allows us to theoretically control the "information trade-offs" of node features and graph topology. Moreover, it can also generate homophilic and heterophilic graphs with the same "amount of information" in their graph topology. This is another contribution that should not be overlooked.
>
> 2.	"page 2: you keep using the term "large step propagation" several times, without actually defining what that means."
>
> What we mean is propagation for a large number of steps (for example, $H^{(k)}$ for large $k$ in GPR-GNN). We agree with the reviewer that this should be clarified in the paper and will address it in the revised paper.
>
> 3.	"Thm 4.1: is there a formal version of this theorem with a proof? if not it's probably not fair to call it a theorem."
>
> As we clearly pointed out in the introduction, all proofs and formal theorem statements are relegated to the Supplement due to space limitations. The formal statement of Theorem 4.1 and its proof can be found in Section A.3 and Theorem 4.2 in Section A.4 in the Supplement. Please refer to this part of the text to convince yourself that we did not call a result a theorem without formally justifying it.
>
> 4.	"please make the text in figures 3 and 4 larger. It's hard to read."
>
> We thank the reviewer for his/her useful suggestion. We will make the figures more readable in the revised paper, if space constraints allow it. Note that all figures in Fig.3 and Fig.4 (a)-(d) have a larger and easier-to-view version in the Supplement.

---

### Official Review · AnonReviewer3 · 2020-10-29
**Great paper, enjoyed reading it**

**Rating:** 9
**Confidence:** 4

**Review:**


The paper proposes a new neural net architecture based partially on the previously proposed Generalized
PageRank (GPR). The model adaptively learns the GPR weights so as to jointly optimize node feature and topological information extraction. The main advantage of  this approach is that -- unlike previously proposed GCNs -- this approach works well for both homophilic and heterophilic graphs (due to the use of GPR). This allows for e.g. node classification without a priori knowledge about the type of graph at hand. Additionally, this approach avoids feature over-smoothing, a known problem in GCNs.

############

I recommend this paper for acceptance due to its novelty and its algorithmic contribution. I really enjoyed reading it.

############

Pros
+ A very interesting approach with useful practical implications for a node classification of both both homophilic and heterophilic graphs.
+ Very clearly written and well articulated.
+ Great literature review of related topics in the area of GCNs.

############

Cons/Suggestions
- I found the explanation of GPR in the beginning of Sec 3 a bit confusing. Please rewrite to add a bit more detail (e.g. what is gamma?), I had to consult the original  paper to understand the notation here.

---

> ### Author Response · Authors · 2020-11-14
> **A response to the suggestion from reviewer 3**
>
> We thank Reviewer 3 for her/his valuable feedback and recommendations for improving the manuscript. The only concern raised is addressed below.
>
> 1.	"I found the explanation of GPR in the beginning of Sec 3 a bit confusing. Please rewrite to add a bit more detail (e.g. what is gamma?), I had to consult the original paper to understand the notation here."
>
> We apologize for creating confusion. Due to space limitations, we tried to make our paper as concise as possible and may have therefore omitted some important explanations. We will add these to Section 3 of our revision, in which one additional page is allowed.

---

### Official Review · AnonReviewer1 · 2020-10-30
**The paper combines GNN with Generalized PageRank to handle two significant weakness of most GNNs, but lack of some important baselines and discussions**

**Rating:** 7
**Confidence:** 3

**Review:**

The paper proposes a new GNN architecture based on Generalized PageRank to handle two weakness in some existing GNNs: the difficulty of neighborhood aggregation on heterophilic graphs, and the oversmoothing problem when stacking GNN layers. The proposed GPR-GNN can be viewed as an extension of the Personalized PageRank-based GNNs, such as APPNP and SGC, which also aimed to handle oversmoothing problem. Pros:  The paper highlights and clearly explains why optimizing the proposed layerwise GPR weights in GNN could tackle the two challenges. Although the final GPR-GNN architecture is simple, the ideas behind are significant. The paper is technically sound. The paper clearly analyzes the differences and relationships between related prior works and the proposed approach. In experiment, datasets are sufficient, including both homophilic and heterophilic graphs (both synthetic datasets and 10 real benchmarks are sufficient) which proves that GPR-GNN can achieve state-of-the-art performances especially on heterophilic graphs.
Cons:
* Baselines: The paper compares GPR-GNN with the state-of-the-art APPNP, a Personalized PageRank-based GNN. However, it also mentions SGC as another most related work based on PPR, but does not use it as a baseline.
* Since the GPR-GNN only have one set of NN parameter \theta in the first layer, and other layers do not have such feature transformation parameters, maybe add more efficiency analysis would be better, since APPNP also has efficiency analysis.
* There are five heterophilic benchmarks. The performances of GPR-GNN on Chameleon and Squirrel are impressive, but GPR-GNN actually does not "significantly outperform" APPNP on Actor, Texas and Cornell datasets, which contradicts the claims. More discussions about this would be better. If it is due to other statistic differences between heterophilic benchmarks, it is necessary to mention them in Table 1.

---

> ### Author Response · Authors · 2020-11-14
> **A few remarks on the points raised by the reviewer**
>
> We thank Reviewer 1 for her/his valuable feedback and recommendations for improving the manuscript. We addressed all the concerns as described below.
>
> 1. "The paper compares GPR-GNN with the state-of-the-art APPNP, a Personalized PageRank-based GNN. However, it also mentions SGC as another most related work based on PPR, but does not use it as a baseline"
>
> We did not include result on SGC in the text as the method performs worse than APPNP in most cases we tested. Nevertheless, we will include the results we obtained for SGC back into our revision. For completeness, we will also include the results for SAGE.
>
> 2.	"Since the GPR-GNN only have one set of NN parameter $\theta$ in the first layer, and other layers do not have such feature transformation parameters, maybe add more efficiency analysis would be better, since APPNP also has efficiency analysis."
>
> We thank reviewer 1 for her/his insightful suggestion. We agree that adding an efficiency analysis similar to that done for APPNP can further strengthen the results in our manuscript and will do so. As a quick answer, it is intuitively clear that GPR-GNN has approximately the same computational complexity as APPNP. We only need to learn $K+1$ additional parameters (the GPR weights $\gamma_k,k = 0,1,\cdots,K$) in our GPR-GNN framework. That is, compared to APPNP with the same $K$ and neural network component, our GPR-GNN will merely require (K+1) additional gradient computations. In practice, $K$ is much smaller than the number of parameters in the NN part. For example, in our experiments we used a 2-layer MLP with $= 64$ hidden units, which is also the default setting for APPNP. We will also include the empirical training time in our revision.
>
> 3.	"There are five heterophilic benchmarks. The performances of GPR-GNN on Chameleon and Squirrel are impressive, but GPR-GNN actually does not "significantly outperform" APPNP on Actor, Texas and Cornell datasets, which contradicts the claims. More discussions about this would be better. If it is due to other statistic differences between heterophilic benchmarks, it is necessary to mention them in Table 1."
>
> We thank the reviewer for her/his comment but would like to point out that for the Actor and Texas dataset, unlike stated by the reviewer, GPR-GNN still offers better performance then APPNP with statistical significance. Nevertheless, we agree that Actor, Texas and Cornell datasets seem to be quiet different from the Chameleon and Squirrel datasets from a statistical point of view. Unfortunately, this difference is not something that can be accurately characterized solely by the homophily measure $\mathcal{H}(G)$ (proposed by Pei et al. [1]). A much more in-depth study is needed to properly capture those differences and this falls out of the scope of the current submission.
>
> To illustrate the above point, consider the case where we have $4$ classes with labels $y_i\in \\{ 0,1,2,3 \\}$ and two graph structures in which: 1) Nodes with label $1$ are linked to all nodes with label $3$ and $4$. Nodes with label $2$ are also linked to all nodes with label $3$ and $4$.; 2) nodes with label $1$ are linked to nodes with label $3$ and nodes with label $2$ are linked to nodes with label $4$. By definition, both cases induce graphs with $\mathcal{H}(G) = 0$. However, as is intuitively clear, the graph topology from case 2) carries more information about the underlying cluster structures. Currently we do not have a better definition for homophily measures available but this is indeed an interesting topic to study in the future. We will include this discussion along with the illustrative example presented into our revision.
>
> References
>
> [1] Hongbin Pei, Bingzhe Wei, Kevin Chen-Chuan Chang, Yu Lei, and Bo Yang. Geom-gcn: Geometric graph convolutional networks. In International Conference on Learning Representations, 2019.

---

### Official Review · AnonReviewer4 · 2020-11-01
**An attempt to incorporate Generalized PageRank into GNNs to cope with heterophily and oversmoothing**

**Rating:** 4
**Confidence:** 4

**Review:**

Paper Summary:

Paper summary:
This paper attempts to incorporate generalized page rank (GPR) in Graph Neural Networks (GNNs), aiming to address two problems plaguing GNNs: i) Existing message-passing based GNNs are best suited to handle graph-structured data with the homophily property and may not be able to handle those with the heterophily property. ii) Existing GNNs suffer over-smoothing which force them to be "shallow." The key idea of GPR is to introduce a weight factor $\gamma_k$ in the kth iteration of computing the page rank score. The authors conduct extensive evaluation using bencmark datasets, and show improvements over some existing GNN methods.

Pros:
  + Attempt in addressing the potential limitations of message-exchange based GNNs that may not be suited to deal with graph datasets with heterophily relations.
  + Evaluation using both synthetic data based on  contextual stochastic block models (cSBM) and real-world "heterophily" datasets.

Concerns:

  - The paper starts by talking about the "trade-offs" between node features and graph topology. However, the bulk of the paper is focused on using GPR as a more "general" graph filter as in the "standard" GNNs. The latter in recent years, unfortunately, since the work of Kipf & Welling, have primarily used variants of a simple graph of the form I+A.  Based on the GPR-GNN model shown in Fig.1, the node features {X_i} are only used to initialize H^{(0)}. In other words, the node features do not factor into the choice (or learning) of \gamma_k's in the latter stage.

 - One would assume that where a graph or network is "homophily" or "heterophily" would be in some matter encoded in the node features, apart from the network topology. In other words, the same network topology may appear in both homophily and heterophily datasets. [This might not be true in practice: one might imagine that heterophily datasets might likely contain bi-partite or multi-partite structures than homophily datasets.]

 - From a theoretical point of view, GPR as used in the paper does not really provide a trade-off between node features and graph topology. As a "polynomial graph filter", the parameters {\gamma_k} simply "modulate" the eigenvalues. In particular, if  {\gamma_k} (as an infinite sequence) is convergent,  the series \sum_k \gamma_k --> \gamma, then \sum_k \gamma_k A^k --> (\gamma)(I-A)^{-1}  (assuming A is normalized, the dominant eigenvalue is less 1), A^kH^{(k)} would converge to the eigenvector  (a normalized node degree vector) associated with the dominant eigenvalue, thus independent of the initial vector H^{(0)} that derives from the node features {X_i}.

 - Of course, the paper uses a fixed K power iteration (instead of letting K goes infinity). The issue of oversmoothing comes from when K is large, the "dominant" eigenvector takes over the "graph structure". I am not convinced that GPR helps either address the heterophily or the oversmoothing issues of GNNs. The use of GPR is motivated by the reference [Li et al 2019] "Optimizing Generalized PageRnak Methods for Seed-Expanision Community Detection". However,in [Li et al] GPR is used for community detection, where "homophily" is actually assumed. Note that stochastic block models (SBMs)  are fundamentally graphs with "homopily"  where nodes are likely to connected with nodes within the same community.

 - All in all, GPR is yet another way to compute some "centrality" or ranking among the nodes. It does not fundamentally capture the structural (local or global) properties of graphs. There are other (perhaps "better" metrics, see, e.g., the paper "Hunt For The Unique, Stable, Sparse And Fast Feature Learning On Graphs" (NeurISP 2017), which is used to develop a Graph Capsule GNN. Unfortunately, some of these metrics are more computationally expensive to compute for large graphs.

Other comments:
 - As an aside, one can in fact absorb the parameter \gamma_k as part of the weight vector $W^{(k)}$ to be learned during the  kth-layer of the GNN. I believe that this might be the advantage gained by GPR-GNN, where in many existing GNNs the weight vector $W{(k)}$ is re-used (i.e., the same weight vector is often used in different layers). In GPR-GNN, the learned weight vector is rescaled by \gamma_k.

- The theoretical results in Section 4 in terms of "graph filters"  are largely well known, see, the survey paper by Michael M. Bronstein et al, "Geometric deep learning: going beyond Euclidean data." IEEE Signal Processing Magazine, 2017.

- I am curious about the specific GNNs that you used to compare with your method, besides Geom-GNNs. There are many other GNNs, e.g., GIN, SAGE that are not chosen for comparison.

---

> ### Author Response · Authors · 2020-11-14
> **Response to all concerns raised by the reviewer**
>
> [Response split in three posts: 1/3]
> We thank Reviewer 4 for her/his valuable feedback and recommendations for improving the manuscript. However, we respectfully disagree with most of the critical points raised by the reviewer and hope to convince her/him about the validity of our claims.
>
> 1.	"...Based on the GPR-GNN model shown in Fig.1, the node features ${X_i}$ are only used to initialize $H^{(0)}$. In other words, the node features do not factor into the choice (or learning) of $\gamma_k$'s in the latter stage."
>
> This statement is incorrect. In fact, we did emphasize multiple times that the GPR weights $\gamma_k$s are $\textbf{jointly optimized}$ with the neural network part $f_\theta$, and this statement also features in our abstract. In the description of Figure 1, we once again emphasizes that both GPR weights $\gamma_k$ and the neural network $f_\theta$ are learned simultaneously in an end-to-end fashion. Note that $\gamma_k$s are learned using the gradient of the loss function which obviously depends on the node features $X$. Hence, the claim that the node features $X$ do not affect the learning process for the parameters $\gamma_k$ is incorrect.
>
> 2.	"One would assume that where a graph or network is "homophily" or "heterophily" would be in some matter encoded in the node features, apart from the network topology...."
>
> Note that we used the definition of homophily and heterophily from [1], and this is clearly stated on page 3. By referring to our and prior work, one can see that whether a graph is homophilic or heterophilic depends on \textbf{both} the graph topology and the node labels, but not the node features (see also [2] for the similar definition on homophily and heterophily, which again depends only on the graph topology and node labels but not the node features). Hence, there is no ambiguity in our definition and the statement made by the reviewer 4 is unjustified.
>
> 3.	"From a theoretical point of view, GPR as used in the paper does not really provide a trade-off between node features and graph topology. As a "polynomial graph filter", the parameters {$\gamma_k$} simply "modulate" the eigenvalues. In particular, if {$\gamma_k$} (as an infinite sequence) is convergent, the series $\sum_k \gamma_k \rightarrow \gamma$, then $\sum_k \gamma_k A^k \rightarrow (\gamma)(I-A)^{-1}$ (assuming A is normalized, the dominant eigenvalue is less 1), $A^kH^{(k)}$ would converge to the eigenvector (a normalized node degree vector) associated with the dominant eigenvalue, thus independent of the initial vector $H^{(0)}$ that derives from the node features {$X_i$}."
>
> This comment is incorrect. First, the claim that $\sum_k \gamma_k A^k \rightarrow \gamma (I-A)^{-1}$ is flawed. From a Taylor series expansion, we have that $(I-A)^{-1} = \sum_k A^k$. Under the assumption made by reviewer 4 that $\sum_k \gamma_k = \gamma$, it is obvious that $\gamma(I-A)^{-1} = (\sum_k \gamma_k)( \sum_k A^k)\neq \sum_k \gamma_k A^k$. Hence, the criticism of our work is based on a wrong argument. Second, there is no term $A^kH^{(k)}$ even mentioned in our manuscript. By definition $H^{(k)} = \tilde{A}_{sym}^kH^{(0)}$. This is yet another misunderstanding regarding our results.
>
> We believe that what reviewer 4 really tried to say is that $H^{(k)}$ becomes independent of the initial vector $H^{(0)}$ for $k$ sufficiently large. This is exactly the over-smoothing problem in GCN which comes from only using the last step propagation results. In contrast, we aggregate results from all steps of propagation with GPR weights $Z = \sum_k \gamma_k H^{(k)}$. We have shown in Theorem 4.2 that when the $\gamma_k$s are learnable, GPR-GNN can prevent over-smoothing. Please check the formal statement of Theorem 4.2 in the Supplement for more details.

---

> > ### Author Response · Authors · 2020-11-14
> > **Response to all concerns raised by the reviewer (part 2)**
> >
> > [Response split in three posts: 2/3]
> > Finally, we would like to explain once again why GPR provides trade-off between the use of node features and the graph topology. From the formula $Z = \sum_k \gamma_k H^{(k)}$ we can see that GPR weights $\gamma_k$ control the contribution of each step $H^{(k)}$ in the final output. When we use only $H^{(0)}$, we purely use the node feature information and neglect the graph topological features. On the other hand, if we use merely $H^{(K)}$ for some sufficiently large $K$, we only use (but not optimally) the graph topological features as $H^{(K)}$ is nearly independent of $H^{(0)}$ (and thus $X$). Hence, the GPR weights $\gamma_k$ indeed ensure a trade-off between node feature and graph topology as they control the combination of $H^{(k)}$ in an adaptive manner. Furthermore, we showed through experiments on cSBM that our GPR-GNN can indeed offer good trade-off. In Figure 2, when the graph topology is uncorrelated with the node labels ($\phi = 0$ case), GPRGNN has similar performance as MLP. Correspondingly, in this case $\gamma_0$ has the largest magnitude among the learnt GPR weights (Figure 6 in the Supplement). For the case where node features are uncorrelated with the node labels ($\phi = \pm 1$), we can again observe that GPR-GNN has better performance compared to baseline methods. The learnt GPR weights are again explainable (Figure 6 in the Supplement).
> >
> > 4.	"Of course, the paper uses a fixed K power iteration (instead of letting K goes infinity). The issue of oversmoothing comes from when K is large, the "dominant" eigenvector takes over the "graph structure". I am not convinced that GPR helps either address the heterophily or the oversmoothing issues of GNNs."
> >
> > Note that we proved in Theorem 4.2 that indeed our GPR-GNN can prevent over-smoothing. Please also check our response to other reviewers as well. We also provide extensive experiments (Figure 4 (e)-(f), Table 7 and Figure 8 in the Supplement) to support our theoretical results. In our experiments, we use $K=10$ which is significantly larger than the number of layers in standard GCN and GAT, in which $2$-$4$ layers are used.
> >
> > Regarding the reason that GPR can address heterophilic data, we have clearly explained it on page 3 (through the equivalence of GPR and polynomial graph filtering) and in Section 4, page 5. Note that it is obvious that $\sum_{k=0}^K \gamma_k\tilde{A}_{{sym}}^k$ is a polynomial graph filter of order $K$ as stated in Section 4. It is well-known that we can approximate any graph filter using a polynomial graph filter [3], which we already mentioned on page 3. Furthermore, we have shown in Theorem 4.1 that $\gamma_k = (-\alpha)^k,\alpha\in (0,1)$ corresponds to a high-pass graph filter and thus GPR-GNN can indeed take account for heterophilic graph data. Our theoretical results match our learnt GPR weights in the experimental section. See Figure 3 for the experiment on cSBM and Figure 4 for the learnt GPR weights on the real-world dataset.
> >
> > 5.	"The use of GPR is motivated by the reference [Li et al 2019] "Optimizing Generalized PageRnak Methods for Seed-Expanision Community Detection". However, in [Li et al] GPR is used for community detection, where "homophily" is actually assumed. Note that stochastic block models (SBMs) are fundamentally graphs with "homopily" where nodes are likely to connected with nodes within the same community."
> >
> > Note that the authors of [4] derived the optimal GPR weights for the SBM which they called IPR, which is just one special case of GPR. Despite the fact that IPR is tailored for homophilic graphs as we already mentioned in page 3, GPR methods in general do not have to restrict themselves to homophilic or heterophilic graphs. Please see the relation of GPR with polynomial graph filitering on page 3 or in our response above. Thus, the claim that GPR cannot deal with heterophilic graphs is incorrect. Also, note that we are using a contextual SBM in our evaluations and not the standard SBM as the former allows for modeling both heterophilic and homophilic data.

---

> > > ### Author Response · Authors · 2020-11-14
> > > **Response to all concerns raised by the reviewer (part 3)**
> > >
> > > [Response split in three posts: 3/3]
> > >
> > > 6.	"All in all, GPR is yet another way to compute some "centrality" or ranking among the nodes. It does not fundamentally capture the structural (local or global) properties of graphs..."
> > >
> > > This comment is also unfounded. First, we have shown that GPRs directly relate to polynomial graph filtering. For $K$ large enough, the optimal GPR can approximate any graph filter arbitrarily well. Thus, GPR can indeed capture the structural properties of graphs. The authors of [4] showed that IPR (the optimal GPR weight derived for SBM) has approximately the same clustering performance as belief propagation (BP) on SBM. It is well known that BP is statistically optimal for addressing SBM problems [5]. This is another piece of evidence that GPR is able to capture the graph topology information accurately.
> > >
> > > As a remark, the centrality score does capture structural information of graphs. In fact, there is a definition of graph centrality based on the eigenvector of the underlying adjacency matrix, which generalizes to hypergraphs as well [6]. Hence, we are quite confused with the claim that the centrality score does not fundamentally capture the topological information of graphs.
> > >
> > > 7.	"The theoretical results in Section 4 in terms of "graph filters" are largely well known, see, the survey paper by Michael M. Bronstein et al, "Geometric deep learning: going beyond Euclidean data." IEEE Signal Processing Magazine, 2017."
> > >
> > > We strongly disagree with this comment. Although the idea of graph filter is well-known (as we pointed out by citing [3]), the theoretical results that we derived in Section 4 are novel and not described in Bronstein's work. For example, we show how GPR-GNN can prevent over-smoothing in Theorem 4.2. We provide concrete examples on what kind of GPR weights $\gamma_k$ will correspond to low-pass and high-pass graph filters in Theorem 4.1. None of these theoretical results appear in the paper provided by reviewer 4.
> > >
> > > 8.	"As an aside, one can in fact absorb the parameter $\gamma_k$ as part of the weight vector $W^{(k)}$ to be learned during the kth-layer of the GNN. I believe that this might be the advantage gained by GPR-GNN, where in many existing GNNs the weight vector $W^{(k)}$ is re-used (i.e., the same weight vector is often used in different layers). In GPR-GNN, the learned weight vector is rescaled by $\gamma_k$."
> > >
> > > We show via extensive experiments that our GPR-GNN significantly outperforms GCN and GAT on heterophilic datasets. Also, GCN and GAT are known to suffer from the over-smoothing problem. To the best of our knowledge, there is no work on re-using $W^{(k)}$ to provably resolve the over-smoothing problem and deal with heterophilic data simultaneously. It would be of great value to the authors if Reviewer 4 can provide a concrete set of references to those lines of work.
> > >
> > >
> > > References
> > >
> > > [1] Hongbin Pei, Bingzhe Wei, Kevin Chen-Chuan Chang, Yu Lei, and Bo Yang. Geom-gcn: Geometric graph convolutional networks. In International Conference on Learning Representations, 2019.
> > >
> > > [2] Jiong Zhu, Yujun Yan, Lingxiao Zhao, Mark Heimann, Leman Akoglu, and Danai Koutra. Generalizing graph neural networks beyond homophily. NeurIPS 2020.
> > >
> > > [3] D. I. Shuman, S. K. Narang, P. Frossard, A. Ortega, and P. Vandergheynst. The emerging field of signal processing on graphs: Extending high-dimensional data analysis to networks and other irregular domains. IEEE Signal Processing Magazine, 30(3):83–98, 2013.
> > >
> > > [4] Pan Li, I Chien, and Olgica Milenkovic. Optimizing generalized pagerank methods for seedexpansion community detection. In Advances in Neural Information Processing Systems, pp. 11705–11716, 2019.
> > >
> > > [5] Emmanuel Abbe. Community detection and stochastic block models: recent developments. The Journal of Machine Learning Research, pp. 6446—6531, 2017.
> > >
> > > [6] Austin Benson. Three hypergraph eigenvector centralities. SIAM Journal on Mathematics of Data Science, pp. 293-312, 2019.
> > >
> > > [7] Johannes Klicpera, Aleksandar Bojchevski, and Stephan Gunnemann. Predict then propagate: Graph neural networks meet personalized pagerank. In International Conference on Learning
> > > Representations, 2018.

---

### Author Response · Authors · 2020-11-22
**Revision of our manuscript**

We again thank every reviewers for their constructive suggestions on improving our manuscript. We add more experiments and clarifications to address reviewers' concerns. Also, we add Appendix originally from the supplementary material in our revision. Detail changes are as follows:

1. Explanation on "large-step propagation" (suggested by reviewer 2).

2. More explanation on GPR (suggested by reviewer 3).

3. New experiments on SGC and SAGE (suggested by reviewer 1 and 4 respectively).

4. Enlarge text font for fig 3 and 4 (suggested by reviewer 2).

5. More discussion on heterophily dataset (suggested by reviewer 1).

6. Efficiency analysis (suggested by reviewer 1).

7. Add Appendix originally from the supplementary material.

8. Correct some typos.

9. Move the conclusion section back to main text from Appendix.

---

### Decision · Program_Chairs · 2021-01-07
**Final Decision**

**Decision:**

Accept (Poster)

**Comment:**

In this paper, the authors propose a new GNN architecture based on Generalized PageRank to handle two weaknesses in some existing GNNs. The novelty of this approach is that it works well for both homophilic and heterophilic graphs (due to the use of GPR).

Overall the paper is interesting and well written. Moreover,  the authors addressed the concerns of reviewers during the rebuttal period. Thus, I vote for acceptance.